# Real-Time Metaheuristic Algorithm for Dynamic Fuzzification, De-Fuzzification and Fuzzy Reasoning Processes

**Hasitha R. Jayetileke** [1] , **W. R. de Mel** [1] **and Subhas Chandra Mukhopadhyay** [2,*]

1   Department of Materials and Mechanical Technology, Faculty of Technology, University of Sri
    Jayewardenepura, Homagama 10200, Sri Lanka
2   School of Engineering, Macquarie University, Sydney, NSW 2109, Australia
*   Correspondence: subhas.mukhopadhyay@mq.edu.au; Tel.: +61-2-9850-6510; Fax: +61-2-9850-9128

**Abstract:** This paper presents a systematic approach to designing a dynamic metaheuristic fuzzy logic controller (FLC) to control a piece of non-linear plant. The developed controller is a multiple-input–multiple-output (MIMO) system. However, with the proposed control mechanism is possible to adapt it to single-input–single-output (SISO) systems as well. During real-time operation, the dynamic behavior of the proposed fuzzy controller is influenced by a metaheuristic particle swarm optimization (PSO) mechanism. Nevertheless, to analyze the performance of the developed dynamic metaheuristic FLC as a piece of non-linear plant, a 1 kW four-wheel independent-drive electric rover is controlled under different road constraints. The test results show that the proposed dynamic metaheuristic FLC maintains the wheel slip ratio of all four wheels to less than 0.35 and a top recorded translational speed of 90 km/h is maintained for a fixed orientation.

**Keywords:** fuzzy logic (FL); particle swarm optimization (PSO); brushless sensored direct-current motor control; four-wheel independent-drive

## 1. Introduction

Contemporary science, engineering and technology are based on mathematical models, which are derived through system-governing equations and algorithmic computer programs [1,2]. Therefore, an appropriate mathematical model needs to be established with respect to the physical parameters when analyzing or controlling the behavior of a system. When modelling physical systems, if the information related to the system is continuous then differential equations [3] need to be derived, and if not that means that if the information is discrete then difference equations need to be derived. Moreover, when the information related to the governing equation(s) is continuous and if more than one dependent variable or parameter describes the system behavior, then partial differential equations need to be established; in such cases, this information or the system parameters that relate to measurable physical quantities are needed. To analyze or control a system that is described by such mathematical equations, a mechanism or algorithmic approach is needed to find an appropriate solution for such system-governing equations.

However, when the system behavior becomes non-linear, there will be problems that cannot be solved using this approach. Therefore, non-linear system models need to be considered and the non-linear equations need to be sophisticated. In practice, the answers to such challenges may often be found despite the lack of an appropriate model or an algorithmically acceptable model. This is because in such a mathematical model the information is not precise, meaning it will have fuzziness.

Fuzzy logic is a mathematical theory that is based on fuzzification, defuzzification, fuzzy inference and fuzzy-rule-based mechanisms. Therefore, fuzzy logic is capable of dealing with imprecise information, where such problems have no ordinary mathematical model or acceptable algorithmic solution [4]. Such problems are often encountered in

automobile engineering or technology when controlling the wheel slip of each wheel (independently), as an example of such a complex phenomenon.

The functioning of fuzzy logic, such as fuzzification, defuzzification, fuzzy inference and fuzzy-rule-based mechanisms, is based on two main concepts, which are the fuzzy set and fuzzy linguistic variables. These concepts could be illustrated by considering the translational velocity (speed) of a vehicle. The vehicle speed is a measurable quantity; however, in practice, it is often stated that the speed is slow, moderate or fast rather than giving an exact speed amount in km/h or mph. With respect to fuzzy logic concepts, speed is a linguistic variable that can be assigned different values for slow, moderate and fast. For example, speeds between 40 and 70 km/h ($\approx 25$ to $\approx 40$ mph) are considered moderate, while speeds lower than 40 km/h are slow and higher than 70 km/h are fast. In accordance with the ordinary mathematical approach, it is possible to say that there are three groups (sets) of speed values, namely a set of slow values (all values below 40 km/h), a set of moderate values (all values between 40 to 70 km/h) and a set of fast values (all values above 70 km/h). Thus, a speed of 39.9 km/h would be slow but 40.1 km/h would be moderate. Such sets have rigid (sharp) boundaries. However, boundaries, in reality, are not rigid. For example, a moderate speed for one person may be excessively slow for another. This gives rise to the idea of a fuzzy set, which is defined by a membership function that can have any value between (and including) 0 and 1. In such cases, this membership function corresponds to a membership value for every speed value. This membership value is also known as the grade of membership or the degree of membership. Moreover, there may be situations that occur where the membership functions will overlap with neighboring fuzzy sets. Therefore, the values of the crisp variable speed may contain a value belonging to all merged fuzzy sets. However, all of the defined membership functions (fuzzy sets) create a quantitative relationship between the linguistic variables and conventional quantifiable quantity. In a SISO/MIMO fuzzy system, with respect to input–output fuzzy linguistic variables, if appropriate fuzzy rules are defined, such a fuzzy controller could be used to control physical plants. In such a fuzzy logic controller (FLC), at the input side the ordinary or conventional variables represent the physical parameters (such as the speed). The first phase, termed fuzzification, evaluates all membership functions and finds membership values or membership grades that correlate with the values of the input variables. Subsequently, "if" and "then" operations are executed in the second phase (also known as fuzzy inference). The outcomes of these processes define the form or the shape of the fuzzy set membership function applied to the output linguistic variable. In the final phase, the membership function of the output fuzzy set is used to discover the most appropriate crisp value of the output variable, which is known as defuzzification.

To control a plant related to a physical phenomenon via fuzzy logic, one has to define the input–output variables, fuzzy sets or fuzzy membership functions and fuzzy rules. Generally, these types of fuzzy logic systems (FLS) are known as type-1 FLS/FLCs. Table 1 shows the current state of this research field, with key publications showing evidence of the sophisticated development of fuzzy input–output membership functions (shown in the 4th and 5th columns), and especially the fuzzy inference mechanism (FIM) due to the sophisticated physical phenomena. Moreover, in some situations, the definitions of the fuzzy sets and fuzzy rules are not obvious. Table 1 shows that for an FLC to become more realistic, hundreds of fuzzy rules have to be implemented (as shown in the 6th column) in the FIM. Even in such a system, the defined input–output fuzzy sets will remain as they are. Therefore, developing such a controller becomes more tedious and time-consuming when the system becomes non-linear (uncertain). A previous study [5] showed that even after developing such a complicated FIM, tuning the fuzzy rules is also a crucial issue.

**Table 1.** A general summary of the current state of the research area that shows the sophisticated development of fuzzy input–output membership functions (shown in the 4th and 5th columns), especially when developing the FIM (the 6th column shows evidence of the development of a large number of fuzzy rules) to compensate for sophisticated physical phenomena.

| Ref. | Year | Contribution | Number of Input Fuzzy Membership Functions | Number of Output Fuzzy Membership Functions | Number of Fuzzy Rules |
|---|---|---|---|---|---|
| [6] | 2021 | An Agent-Based Model-Driven Decision Support System for Assessment of Agricultural Vulnerability of Sugarcane Facing Climatic Change: Crop yield model (*Scopus-Indexed*) | 8 | 5 | 768 |
| [7] | 2021 | New FMEA Risks Ranking Approach Utilizing Four Fuzzy Logic Systems (*Scopus-Indexed*) | 4 | 1 | 625 |
| [8] | 2021 | Symptom Analysis Using Fuzzy Logic for Detection and Monitoring of COVID-19 Patients (*Scopus-Indexed*) | 6 | 1 | 512 |
| [9] | 2021 | A Fuzzy Logic-Based Cost Modelling System for Recycling Carbon Fibre Reinforced Composites (*Scopus-Indexed*) | 5 | 1 | 243 |
| [10] | 2021 | Enhanced Intelligent Closed Loop Direct Torque and Flux Control of Induction Motor for Standalone Photovoltaic Water Pumping System (*Scopus-Indexed*) | 3 | 1 | 180 |
| [11] | 2021 | SAFEA application design on determining the optimal order quantity of chicken eggs based on fuzzy logic (*Scopus-Indexed*) | 3 | 1 | 144 |
| [12] | 2021 | A fuzzy logic-based approach for evaluating forest ecosystem service provision and biodiversity applied to a case study landscape in Southern Germany (*Scopus-Indexed*) | 5 | 5 | 125 |
| [13] | 2021 | A Fuzzy Logic Model for Early Warning of Algal Blooms in a Tidal-Influenced River (*Scopus-Indexed*) | 3 | 1 | 125 |
| [14] | 2021 | Fuzzy Logic in Aircraft Onboard Systems Reliability Evaluation: A New Approach (*Scopus-Indexed*) | 3 | 1 | 125 |
| [6] | 2021 | An Agent-Based Model-Driven Decision Support System for Assessment of Agricultural Vulnerability of Sugarcane Facing Climatic Change: Crop vulnerability model (*Scopus-Indexed*) | 5 | 3 | 120 |
| [15] | 2021 | Inverter current control for reactive power compensation in solar grid system using Self-Tune Fuzzy Logic Controller (*Scopus-Indexed*) | 2 | 1 | 91 |
| [16] | 2021 | A Fuzzy Logic Model for the Analysis of Ultrasonic Vibration Assisted Turning and Conventional Turning of Ti-Based Alloy (*Scopus-Indexed*) | 4 | 4 | 81 |
| [17] | 2021 | Fuzzy Logic Based Synchronization Method for Solar Powered High Frequency On-Board Grid (*Scopus-Indexed*) | 2 | 1 | 81 |
| [6] | 2021 | An Agent-Based Model-Driven Decision Support System for Assessment of Agricultural Vulnerability of Sugarcane Facing Climatic Change: Uncertain parameters model (*Scopus-Indexed*) | 5 | 3 | 72 |

**Table 1.** *Cont.*

| Ref. | Year | Contribution | Number of Input Fuzzy Membership Functions | Number of Output Fuzzy Membership Functions | Number of Fuzzy Rules |
|------|------|--------------|--------------------------------------------|---------------------------------------------|-----------------------|
| [6] | 2021 | An Agent-Based Model-Driven Decision Support System for Assessment of Agricultural Vulnerability of Sugarcane Facing Climatic Change: Non-nutritional disorders model (*Scopus-Indexed*) | 5 | 3 | 72 |
| [18] | 2021 | Prediction of gas velocity in two-phase flow using developed fuzzy logic system with differential evolution algorithm (*Scopus-Indexed*) | 3 | 1 | 64 |
| [19] | 2021 | Comprehensive Knowledge-Driven AI System for Air Classification Process (*Scopus-Indexed*) | 5 | 3 | 55 |
| [20] | 2021 | Overall fuzzy logic control strategy of direct driven PMSG wind turbine connected to grid (*Scopus-Indexed*) | 2 | 1 | 49 |
| [21] | 2021 | Optimal Geno-Fuzzy Lateral Control of Powered Parachute Flying Vehicles (*Scopus-Indexed*) | 2 | 1 | 49 |
| [22] | 2021 | Fuzzy Mathematics-Based Outer-Loop Control Method for Converter-Connected Distributed Generation and Storage Devices in Micro-Grids (*Scopus-Indexed*) | 2 | 1 | 49 |
| [23] | 2021 | A Novel Fuzzy PI Control Method for Variable Frequency Brushless Synchronous Generators (*Scopus-Indexed*) | 2 | 1 | 49 |
| [24] | 2021 | A Fuzzy Multi-Criteria Model for Municipal Waste Treatment Systems Evaluation including Energy Recovery: Workstation evaluation (*Scopus-Indexed*) | 2 | 1 | 49 |
| [24] | 2021 | A Fuzzy Multi-Criteria Model for Municipal Waste Treatment Systems Evaluation including Energy Recovery: Treatment system evaluation (*Scopus-Indexed*) | 4 | 1 | 49 |
| [25] | 2021 | A Temperature Control Method for Micro-accelerometer Chips Based on Genetic Algorithm and Fuzzy PID Control (*Scopus-Indexed*) | 2 | 1 | 49 |
| [26] | 2022 | Induction Motor DTC Performance Improvement by Inserting Fuzzy Logic Controllers and Twelve-Sector Neural Network Switching Table (*Scopus-Indexed*) | 7 | 7 | 49 |
| [27] | 2022 | Fuzzy Hysteresis Current Controller for Power Quality Enhancement in Renewable Energy Integrated Clusters (*Scopus-Indexed*) | 7 | 7 | 49 |
| [28] | 2021 | Fuzzy Logic-Based Controller for Bipedal Robot (*Scopus-Indexed*) | 2 | 1 | 30 |
| [29] | 2021 | A Swarm Intelligence Graph-Based Pathfinding Algorithm Based on Fuzzy Logic (SIGPAF): A Case Study on Unmanned Surface Vehicle Multi-Objective Path Planning (*Scopus-Indexed*) | 3 | 1 | 27 |
| [30] | 2021 | Fuzzy Logic and Modified Butterfly Optimization with Efficient Fault Detection and Recovery Mechanisms for Secured Fault-Tolerant Routing in Wireless Sensor Networks (*Scopus-Indexed*) | 3 | 1 | 27 |

**Table 1.** *Cont.*

| Ref. | Year | Contribution | Number of Input Fuzzy Membership Functions | Number of Output Fuzzy Membership Functions | Number of Fuzzy Rules |
|---|---|---|---|---|---|
| [31] | 2021 | Optimal Routing Protocol for Wireless Sensor Network Using Genetic Fuzzy Logic System (*Scopus-Indexed*) | 3 | 1 | 27 |
| [32] | 2021 | GPS Data Correction Based on Fuzzy Logic for Tracking Land Vehicles: Fuzzy system 1 (*Scopus-Indexed*) | 2 | 1 | 25 |
| [33] | 2021 | Lifting and stabilizing of two-wheeled wheelchair system using interval type-2 fuzzy logic control based spiral dynamic algorithm (*Scopus-Indexed*) | 2 | 1 | 25 |
| [34] | 2021 | Optimization of Fuzzy Logic Based Virtual Pilot for Wargaming (*Scopus-Indexed*) | 2 | 1 | 25 |
| [35] | 2021 | LQR and Fuzzy Logic Control for the Three-Area Power System (*Scopus-Indexed*) | 5 | 5 | 25 |
| [36] | 2021 | Smart Homes as Enablers for Depression Pre-Diagnosis Using PHQ-9 on HMI through Fuzzy Logic Decision System (*Scopus-Indexed*) | 2 | 1 | 20 |
| [37] | 2021 | Pineapple maturity classifier using image processing and fuzzy logic (*Scopus-Indexed*) | 3 | 1 | 18 |
| [38] | 2021 | Algorithm for Preventing the Spread of COVID-19 in Airports and Air Routes by Applying Fuzzy Logic and a Markov Chain (*Scopus-Indexed*) | 4 | 1 | 14 |
| [39] | 2021 | Artificial Intelligence Search Strategies for Autonomous Underwater Vehicles Applied for Submarine Groundwater Discharge Site Investigation (*Scopus-Indexed*) | 3 | 2 | 13 |
| [40] | 2021 | Intelligent Fault Detection and Identification Approach for Analog Electronic Circuits Based on Fuzzy Logic Classifier (*Scopus-Indexed*) | 3 | 1 | 12 |
| [32] | 2021 | GPS Data Correction Based on Fuzzy Logic for Tracking Land Vehicles: Fuzzy system 2 (*Scopus-Indexed*) | 2 | 1 | 9 |
| [41] | 2021 | Optimum Design of a Composite Optical Receiver by Taguchi and Fuzzy Logic Methods (*Scopus-Indexed*) | 3 | 1 | 9 |
| [42] | 2022 | SOC Balancing and Coordinated Control Based on Adaptive Droop Coefficient Algorithm for Energy Storage Units in DC Microgrid (*Scopus-Indexed*) | 3 | 3 | 9 |
| [43] | 2021 | Fuzzy Logic in Selection of Maritime Search and Rescue Units (*Scopus-Indexed*) | 2 | 1 | 6 |

However, the level of uncertainty in a system can be minimized by employing interval type-2 fuzzy logic, which has stronger capabilities to handle uncertainties by modelling the vagueness and unpredictability of information [44–47]. This is because the growth of type-2 FLS uncertainty can be directly integrated into fuzzy sets, as described in Section 6. Furthermore, in the last three years of studies on higher-order types of FLS in particular, the designed and developed applications of interval type-2 fuzzy logic have increased significantly [48–54]. These type-2-based FLS applications have been identified in artificial intelligence (AI) [55–59], adaptive control [60–66], electric motor control [67–72], Internet of Things (IoT) [73–77], digital image processing [78–84] and other areas [85–87]. Of

course, the application of interval type-2 fuzzy logic in the domain of control has recently attracted a lot of attention due to its better performance under uncertain conditions. The fundamental issue, however, is the complexity of designing and constructing interval type-2 fuzzy controllers, which contain more parameters than their type-1 counterparts; therefore, this causes greater computational complexity and overhead issues [88–99]. Therefore, several efforts were made to reduce the complexity of generalized interval type-2 fuzzy logic systems; for example, Samui and Samarjit [100] published a neural network (NN)-based tuning mechanism and Cagri and Tufan [101] developed a differential flatness-based controller, which both enable computation with generalized type-2 FLS (GT2FLS). However, no general design strategy for finding an optimal type-2 fuzzy model has been proposed yet [102].

The main contribution of this paper is the design and development of a robust FLC that enables researchers to rapidly develop more realistic fuzzy controllers. Therefore, to overcome the problems and drawbacks from the abovementioned review of previous studies, the main advantages of this study are as follows:

1. We evaluated how the type-2 FLS are more capable of performing under uncertain conditions and designed and developed a mechanism to integrate them into the proposed type-1 FLC;
2. We designed and developed an adaptive metaheuristic FIM for type-1 FLS to overcome the problems that are currently faced when developing realistic fuzzy rules (as shown in Table 1, column 6);
3. We designed and developed a fuzzification and de-fuzzification mechanism while integrating the features that were abstracted from the type-2 FLS into type-1 FLS;
4. A real-time dynamic metaheuristic algorithm to automatically optimize all of the abovementioned processes related to dynamic fuzzification, de-fuzzification and fuzzy reasoning was designed and developed;
5. To examine the performance of the proposed controller as a complex physical phenomenon, a four-wheeled independent-drive electric rover was designed and developed to regulate the wheel slip (under high-speed conditions on slippery roads).

## 2. Overall System Design

Figure 1 shows the proposed dynamic metaheuristic FLC implemented with closed-loop control strategies and the master–slave [103] control mechanism. Controller A represents the proposed dynamic metaheuristic FLC. The proposed dynamic metaheuristic FLC is a combination of four identical controllers (the Takagi–Sugeno–Kang particle swarm optimization fuzzy logic controller (TSK-PSO-FLC)), which are separately dedicated to each wheel while synchronizing together in order to regulate the wheel slip. A previous study [104] showed that the designed and developed differential fuzzy logic controller (D-FLC (*controller C*)) was needed to generate the four excitation signals (desired wheel speeds) for each TSK-PSO-FLC, as per the desired throttle amount and the steering angle. Moreover, when the rover travels under high-speed conditions on slippery roads, the change and rate of change of the desired orientation are rectified via the steering fuzzy logic controller (S-FLC (*controller B*)) by continuously monitoring the actual orientation of the rover through the feedback path. Therefore, Figure 1 shows that with respect to the desired steering angle and the actual yaw angle (orientation) of the rover, the S-FLC generates the corrected steering angle and feeds this into the D-FLC. The S-FLC (*controller B*) and the D-FLC (*controller C*) had to be developed due to the nature of the application (electric rover). The modelling of the system dynamics of the rover has been discussed in [104]. However, this paper focuses on the design and development of a dynamic metaheuristic FLC for other research scientists to absorb into their developments.

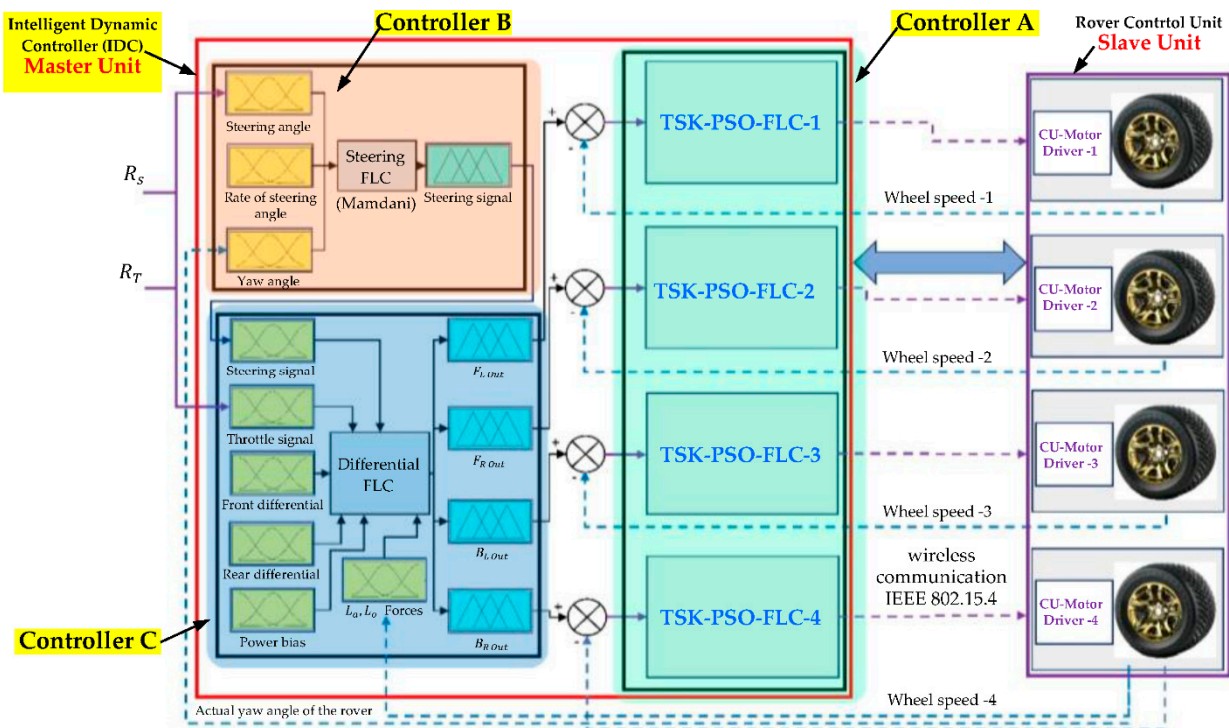

**Figure 1.** Overall block diagram representation of the main control mechanism.

## 3. Implementation of the Dynamic Metaheuristic Fuzzy Logic Controller (TSK-PSO-FLC)

Figure 2 illustrates that the proposed dynamic metaheuristic FLC is implemented in three phases. In phase one, the static Mamdani FLC was developed in a laboratory experimental setup [105] to identify the optimum fuzzy inference engine to compensate for the error and rate of change of the error (motor angular velocity).

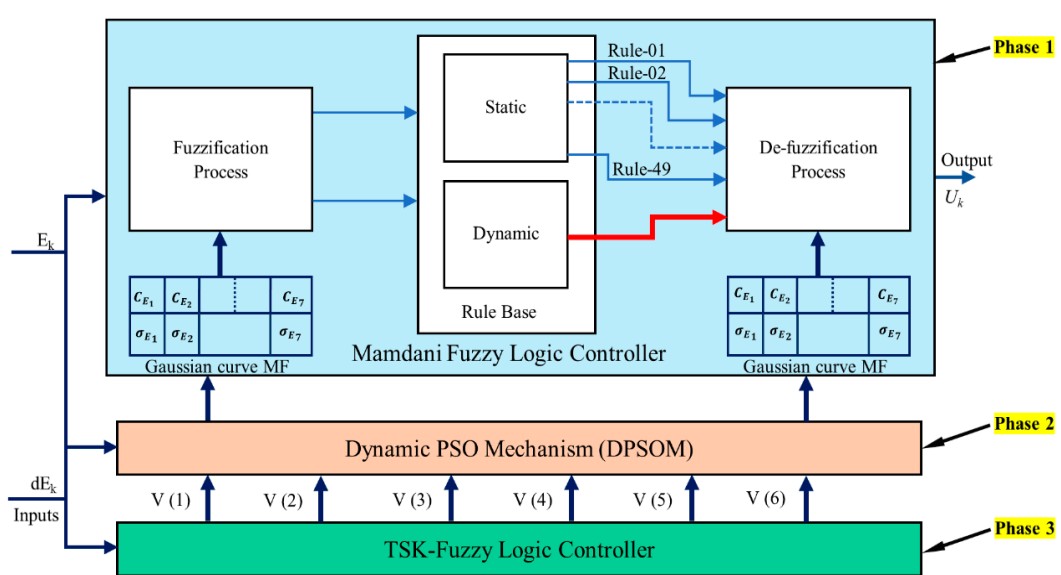

**Figure 2.** Overall block diagram representation of the TSK-PSO-FLC.

## 4. Implementation of the Static Fuzzy Logic Controller (Static FLC)

However, due to the non-linearity behavior of the rover (non-linear wheel slip), a dynamic particle swarm optimization (PSO) mechanism was integrated into the developed static fuzzy controller in phase two (discussed in Section 1). Moreover, in phase three, the

controller had to become more realistic, so a Takagi–Sugeno–Kang (TSK) fuzzy controller was developed to dynamically optimize the parameters of the PSO mechanism. The subsequential Sections 4–6 discuss the development of the static fuzzy controller, the dynamic PSO mechanism and the integrated dynamic PSO mechanism into the developed Mamdani dynamic FLC.

*4.1. Fuzzification Process of the Static Fuzzy Logic Controller*

Figure 3 shows the experimental setup created to test the performance of the proposed static FLC. The proposed static FLC calculates the error $E_k$ and the rate of change of the error $dE_k$ for every discrete sample instant $k$ [105]. Here, $E_k$ and $dE_k$ were computed by the static FLC according to (1) and (2):

$$E_k = (R_k - Y_k) \tag{1}$$

$$dE_k = E_k - E_{(k-1)} \tag{2}$$

where $R_k$ is the desired input trajectory or the reference signal that is proportional to the desired velocity and $Y_k$ is the actual output (actual angular velocity of the motor). These $E_k$ and $dE_k$ crisp values are mapped into the non-singleton fuzzy sets through the fuzzification process. The singleton fuzzy sets are not suitable due to the non-linearity behavior of the DC motor, because of a lack of information about the surroundings. When taking into account a non-singleton fuzzy set, this can be expressed as an elliptic function $\wp$, which means the linguistic variable belonging to that fuzzy set is periodic in two directions (according to $E_k$ and $dE_k$, the linguistic variables can be used to obtain values, either positive or negative). In this case, it contains information about the surroundings (the $E_k$ and $dE_k$ crisp values are affected by uncertainty) and is capable of handling non-linear situations compared to singleton fuzzy sets. This non-singleton fuzzifier for the fuzzy set error $E$ can be expressed as (3):

$$nsg : E \to \wp \ni \forall e \in E \to \mu_e(\cdot) = nsg[e] : \mu_e(e) = 1 \wedge Support[\mu_e(\cdot)] \supset \{e\} \tag{3}$$

where $nsg$ is the non-singleton set, $\forall e$ is the logical operator with a predicate variable $e$ that belongs to the fuzzy set error, $E \to$ is the implication (the if and then condition) and $Support[\mu_e(\cdot)] \supset \{e\}$ is the degree of truth that the implication relation belongs to "$e$". Here, $e$ is an element of $E$. When discrete sampling instants are taken, then (3) can be expressed as for every sampling instant "$k$" as (4):

$$nsg : E_k \to \wp \ni \forall e_k \in E_k \to \mu_{e_k}(\cdot) = nsg[e_k] : \mu_{e_k}(e_k) = 1 \wedge Support\left[\mu_{e_k}(\cdot)\right] \supset \{e_k\} \tag{4}$$

Figure 4 shows that the non-singleton fuzzy sets of the static FLC utilize seven linguistic variables. These variables can be expressed as "NB", "NM", "NS", "ZE", "PS", "PM" and "PB", which mean "*Negative Big*", "*Negative Medium*", "*Negative Small*", "*Zero*", "*Positive Small*", "*Positive Medium*" and "*Positive Big*", respectively.

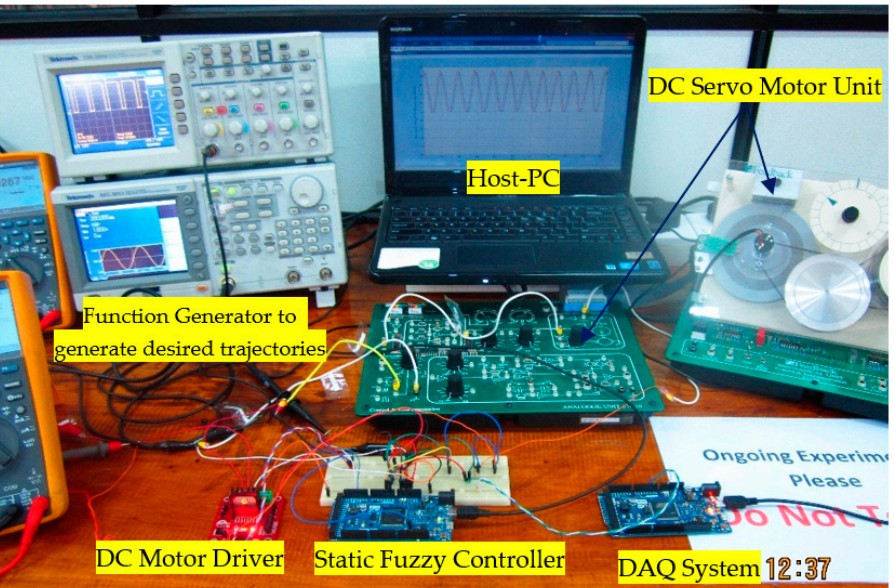

**Figure 3.** The experimental hardware setup of the developed static FLC [105].

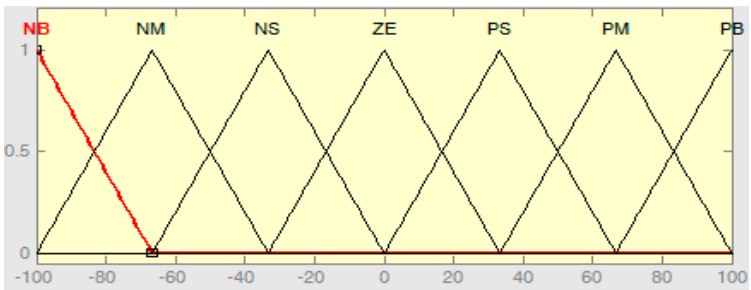

**Figure 4.** The non-singleton fuzzy sets ($E_k$, $dE_k$ and $U$ (Speed)) of the proposed static FLC [105].

### 4.2. Implementation of the Fuzzy Inference Mechanism of the Static Fuzzy Logic Controller

The proposed static FLC is mainly based on the Mamdani fuzzy approach. Through the fuzzification process, $E_k$ and $dE_k$ are fuzzified, and all aforementioned fuzzy sets are passed through the FIM. During this process of designing and developing the fuzzy rule base of the system, the input fuzzy sets belonging to $E_k$ and $dE_k$ are mapped into a different fuzzy set $U_k$ (output fuzzy set (angular speed of the motor) with seven linguistic variables). This fuzzy relation is based on the MAX-MIN composition.

In the fuzzy domain, the linguistic variables are $E$, $dE$ and $U$, with the linguistic values as shown in Figure 4, then the fuzzy sets can be represented as shown in (5)–(7):

$$E = \left\{ \frac{\mu_E(NB)}{NB}, \frac{\mu_E(NM)}{NM}, \frac{\mu_E(NS)}{NS}, \frac{\mu_E(ZE)}{ZE}, \frac{\mu_E(PS)}{PS}, \frac{\mu_E(PM)}{PM}, \frac{\mu_E(PB)}{PB} \right\} \tag{5}$$

$$dE = \left\{ \frac{\mu_{dE}(NB)}{NB}, \frac{\mu_{dE}(NM)}{NM}, \frac{\mu_{dE}(NS)}{NS}, \frac{\mu_{dE}(ZE)}{ZE}, \frac{\mu_{dE}(PS)}{PS}, \frac{\mu_{dE}(PM)}{PM}, \frac{\mu_{dE}(PB)}{PB} \right\} \tag{6}$$

$$U = \left\{ \frac{\mu_U(NB)}{NB}, \frac{\mu_U(NM)}{NM}, \frac{\mu_U(NS)}{NS}, \frac{\mu_U(ZE)}{ZE}, \frac{\mu_U(PS)}{PS}, \frac{\mu_U(PM)}{PM}, \frac{\mu_U(PB)}{PB} \right\} \tag{7}$$

This fuzzy set mapping process is mainly based on the fuzzy-rule-based matrix, and the fuzzy rule base is based on fuzzy propositions (fuzzy statements), which are assigned to the fuzzy sets $E_k$, $dE_k$ and $U_k$. If a fuzzy proposition $\widetilde{P}$ is allocated to a fuzzy set, $\widetilde{A}$, then the truth value $T\left(\widetilde{P}\right)$ of this proposition can be expressed as (8):

$$T\left(\widetilde{P}\right) = \mu_{\widetilde{A}}(x) \tag{8}$$

where $0 \leq \mu \leq 1$.

Therefore, the degree of truth for the proposition $\widetilde{P}$ is that $\widetilde{P} : x \in \widetilde{A}$ is equal to the membership value of $x$ in the fuzzy set $\widetilde{A}$. In FLS, the fuzzy sets in the consequent and antecedent dimensions can make connections with the logical connectives (9), (10), (11) and (12). These connectives are shown by taking into account two propositions $\widetilde{P}$ and $\widetilde{Q}$, which are allocated to the fuzzy sets $\widetilde{A}$ and $\widetilde{B}$, respectively.

Negation (absence):

$$T\left(\overline{\widetilde{P}}\right) = 1 - T\left(\widetilde{P}\right) \tag{9}$$

Disjunction (lack of correspondence): $\widetilde{P} \vee \widetilde{Q} : x$ is $\widetilde{A}$ **or** $\widetilde{B}$, then:

$$T\left(\widetilde{P} \vee \widetilde{Q}\right) = \max\left(T\left(\widetilde{P}\right), T\left(\widetilde{Q}\right)\right) \tag{10}$$

Conjunction: $\widetilde{P} \wedge \widetilde{Q} : x$ is $\widetilde{A}$ **and** $\widetilde{B}$, then:

$$T\left(\widetilde{P} \wedge \widetilde{Q}\right) = \min\left(T\left(\widetilde{P}\right), T\left(\widetilde{Q}\right)\right) \tag{11}$$

Figure 2 shows the rule-based matrix of the developed static FLC based on "if" and "then" statements. This means if the fuzzy proposition $\widetilde{P}$ implies the fuzzy proposition $\widetilde{Q}$, then the fuzzy implication can be expressed as (12).

Implication: $\widetilde{P} \rightarrow \widetilde{Q} : x$ is $\widetilde{A}$, **then** $x$ is $\widetilde{B}$, then:

$$T\left(\widetilde{P} \rightarrow \widetilde{Q}\right) = T\left(\widetilde{P} \vee \widetilde{Q}\right) = \max\left(T\left(\overline{\widetilde{P}}\right), T\left(\widetilde{Q}\right)\right) \tag{12}$$

Then, as in (12), this binary logic implication of the $\widetilde{P}$ and $\widetilde{Q}$ fuzzy sets can be expressed as: $\widetilde{P} \rightarrow \widetilde{Q}$ is **IF** $x$ $\widetilde{A}$, **THEN** $y$ is $\widetilde{B}$; then, this is equivalent to the fuzzy relation $\widetilde{R}$, where $\widetilde{R} = \left(\left(\widetilde{A} \times \widetilde{B}\right) \cup \left(\overline{\widetilde{A}} \times Y\right)\right)$.

Then, the membership function of the fuzzy relation $\widetilde{R}$ can be expressed as in (13).

$$\mu_{\widetilde{R}}(x, y) = \max\left[\left(\mu_{\widetilde{A}}(x)\right) \wedge \mu_{\widetilde{B}}(y), \left(1 - \mu_{\widetilde{A}}(x)\right)\right] \tag{13}$$

In Mamdani-type FLCs, this implication (fuzzy sets connective) can be easily expressed in the fuzzy rule base with the "if" and "then" statement form. The MAX-MIN composition [106] was utilized as the fuzzy relation for this developed static FLC, as mentioned above. If $\widetilde{R}$ is a binary fuzzy relation on $U \times V$ and $\widetilde{S}$ is a binary fuzzy relation on $V \times W$, then the MAX-MIN composition of $\widetilde{R}$ and $\widetilde{S}$ is a binary fuzzy relation on $U \times W$ denoted by $\left(\widetilde{S} \, o \, \widetilde{R}\right)$ and which is given by (14):

$$\left(\widetilde{S}o\widetilde{R}\right)(u, w) = \max\left[\min\left\{\widetilde{R}(u, v)\widetilde{S}(v, w)\right\}\right] \tag{14}$$

For example, the fuzzy relations $\widetilde{R}$ on $U \times V$ and $\widetilde{S}$ on $V \times W$, where $U = \{a, b, c\}$, $V = \{x, y, z\}$ and $W = \{\&, *\}$, are given in the matrix format by:

$$\widetilde{R} = \begin{bmatrix} 1.0 & 0.4 & 0.5 \\ 0.3 & 0.0 & 0.7 \\ 0.6 & 0.8 & 0.2 \end{bmatrix} \quad \text{and} \quad \widetilde{S} = \begin{bmatrix} 0.7 & 0.1 \\ 0.2 & 0.9 \\ 0.8 & 0.4 \end{bmatrix}$$

Considering the elements of $U$, $V$ and $W$, then:

$$
\widetilde{R} \quad = \quad U \begin{array}{c} \\ a \\ b \\ c \end{array} \begin{array}{ccc} \multicolumn{3}{c}{V} \\ x & y & z \\ \hline 1.0 & 0.4 & 0.5 \\ 0.3 & 0.0 & 0.7 \\ 0.6 & 0.8 & 0.2 \end{array}
\qquad\qquad
\widetilde{S} \quad = \quad V \begin{array}{c} \\ x \\ y \\ z \end{array} \begin{array}{cc} \multicolumn{2}{c}{W} \\ \& & * \\ \hline 0.7 & 0.1 \\ 0.2 & 0.9 \\ 0.8 & 0.4 \end{array}
$$

Then, the fuzzy relation $\left(\widetilde{S}o\widetilde{R}\right)$ can be expressed as (14). If we consider the relation for the elements $(a, \&)$ then the fuzzy relation $U \times W$ for $(a, \&)$ can be expressed as (15):

$$
\left(\widetilde{S}o\widetilde{R}\right)(a, \&) = \max\left[\min\left\{\widetilde{R}(a, v), \widetilde{S}(v, \&)\right\}\right] \tag{15}
$$

For every $v$ in $V$, then:

$$
\begin{aligned}
&= \max\left[\min\left\{\widetilde{R}(a, x), \widetilde{S}(x, \&)\right\}, \min\left\{\widetilde{R}(a, y), \widetilde{S}(y, \&)\right\}, \min\left\{\widetilde{R}(a, z), \widetilde{S}(z, \&)\right\}\right] \\
&= \max[\min(1, 0.7), \min(0.4, 0.2), \min(0.5, 0.8)] \\
&= \max[0.7, 0.2, 0.5] \\
&= 0.7
\end{aligned}
$$

According to (15), for the developed static FLC in the inference mechanism, 49 fuzzy relations are utilized. These fuzzy rules are developed based on the experimental test as shown in Figure 3. These "if" and "then" statements are based on fuzzy rules, and the relations can be expressed as follows:

**IF** the ($E_k$ is **NB** AND $dE_k$ is **NB**) **THEN** the $U_k$ is **PB**

All of these developed fuzzy rules, according to the behavior of the DC servomotor, in real-time operation can be formed into a fuzzy-rule-based matrix (7 × 7), as shown in Table 2. The effectiveness of each fuzzy relation or the fuzzy rule depends on the membership grade or the linguistic value of each linguistic variable in each fuzzy set, as mentioned in (5)–(7).

**Table 2.** Fuzzy relations of the developed static FLC [105].

| | | | | | Error ($E_k$) | | | |
|---|---|---|---|---|---|---|---|---|
| | | NB | NM | NS | ZE | PS | PM | PB |
| Rate of Change of Error ($dE_k$) | NB | PB | PB | PB | PS | PS | PS | ZE |
| | NM | PB | PM | PM | PS | PS | ZE | NS |
| | NS | PB | PM | PS | ZE | ZE | NS | NS |
| | ZE | PM | PS | ZE | ZE | ZE | NS | NS |
| | PS | PM | PS | ZE | ZE | NS | NS | NM |
| | PM | PS | PS | NS | NS | NS | NS | NM |
| | PB | ZE | NS | NS | NM | NB | NB | NB |

After the fuzzy relation takes place in the static FLC, it still belongs to a fuzzy set ("speed" ($U_k$) fuzzy set), because during the fuzzification process it will only map the non-linear relation into another non-linear form. According to these fuzzy relations (MAX-MIN composition), to feed the fuzzy decision into the motor drive through a pulse width modulation (PWM) signal, it needs to be converted into a crisp output value. For this purpose, the de-fuzzification process was carried out.

### 4.3. De-Fuzzification Process of the Static Fuzzy Logic Controller

The final goal of this static FLC is to control the brushless direct current (BLDC) motor more precisely and in a stable way to obtain the optimum performance. This task is very sophisticated due to the non-linearity behavior of the wheel slip. However, the fuzzy reasoning mechanism is based on human reasoning. Moreover, in the real world, when developing AI controllers, all of these microprocessors and the embedded systems

are discrete devices that are unable to manipulate fuzzy outputs. Therefore, this fuzzy output needs to be converted into a crisp output that is related to the fuzzy domain. A de-fuzzification mechanism was utilized in this developed static FLC to convert the fuzzy quantity to a precise quantity. The output of this developed static FLC was the logical union of the seven membership functions designed and developed in the universe of discourse of the output variable speed ($U_k$). In this case, the union of these seven memberships functions involved with the MAX operator, as described in Section 4.2, could be expressed as (16):

$$\widetilde{U}_k = \bigcup_{j=1}^{7} \left( \widetilde{U}_{j,k} \right) \tag{16}$$

where $k$ is the sample instant and $j \in \mathbb{Z}^+$ and $1 \leq j \leq 7$.

According to (16), after computing the logical sum to obtain an average output crisp value, the centroid method or the center of gravity method was used as in (17):

$$CoA = \frac{\int_{Xmin}^{Xmax} U(x)x\,dx}{\int_{Xmin}^{Xmax} U(x)\,dx} \tag{17}$$

where $U(x)$ denotes each membership function of the output fuzzy set (angular speed of the DC motor), x denotes the value of the linguistic variable, and $X_{min}$ and $X_{max}$ represent the minimum and maximum ranges of linguistic variables. However, Figure 1 shows that to minimize the rover travel distance (the rover reaction distance for an updated signal received from the controller), the proposed control mechanism should have the capability to respond to a higher frequency for setpoint trajectories. Therefore, Figure 2 shows that to overcome the drawbacks [105] of the developed static FLC, a metaheuristic dynamic FLC is integrated with the static FLC.

## 5. Dynamic Particle Swarm Optimization (PSO) Mechanism

When developing the proposed dynamic metaheuristic FLC for the four-wheeled independent-drive electric rover, mainly 12 non-linear parameters were taken into consideration. These parameters include the throttle level, steering angle, three orthogonal acceleration directions (the longitudinal direction, lateral direction, and radial direction) and three orthogonal angles (such as around the yaw axis, pitch axis and the roll axis). Figure 1 shows that the developed D-FLC [104] took into consideration the behavior of all of these parameters and generated the desired angular velocity and torque required by the proposed dynamic metaheuristic FLC. However, due to the non-linear variations (uncertainties) of the actual angular velocity and torque (because of the non-linear wheel slip), sharp rising and falling edges of the desired input trajectory took place. Therefore, the proposed dynamic metaheuristic FLC should have the ability to quickly generate an optimum solution to enhance the stability. Table 1 shows that under the column "*number of fuzzy rules*", for the FLCs to become more realistic, the research scientists have to strengthen the FIM by creating hundreds of fuzzy rules. Therefore, instead of developing such a sophisticated FIM to overcome these issues, the proposed dynamic metaheuristic FLC is capable of dynamically optimizing the fuzzy reasoning process. Moreover, in addition to optimizing the fuzzy reasoning process, the proposed metaheuristic FLC is capable of dynamically optimizing the fuzzification and de-fuzzification processes as well.

*Implementation of the Dynamic PSO Mechanism*

In this proposed PSO-based optimization mechanism, the main goal is to identify and generate a global minimizer for the objective function (based on $E_k$ and $dE_k$). Many of the optimization mechanisms fulfil this requirement through deterministic methods and probabilistic methods. Deterministic-method-based optimization mechanisms use

heuristic approaches (learning by themselves) by taking into consideration the $E_k$ and $dE_k$ (the error and rate of change of the error). Therefore, the optimization process is controlled through the objective function threshold level, and this objective function is a function of the parameters $E_k$ and $dE_k$. Mostly, the deterministic optimization mechanisms apply penalties to the generated output.

The PSO mechanism is a progressive computation technique based on swarm intelligence that gives a potential solution among the population members in the hyperspace. The optimized solutions are represented by adaptable position change (velocity) particles. The position of each particle in the hyperspace (search space) can be represented by n-dimensional axes. Figure 2 shows that these *coordinate positions* are assigned to dynamically tune the fuzzification process, de-fuzzification process and the FIM. These optimum parameters are the final solution of the PSO controller, which is generated while taking into consideration the personal best $P_{Best}$ of each particle (feasible solutions) until reaching the global best $G_{Best}$ or the social best solution in the search space. These parameters need to be quickly re-optimized because of the dynamic behavior of the wheel slip.

Figure 5 shows the updated positions of the particles that are determined by taking into consideration the updated velocity. When considering the original PSO mechanism [107], in hyperspace if the position of the particle "*i*" is denoted by $x_i(t)$ at a discrete time *t* then the updated position $x_i(t+1)$ and the updated velocity $v_i(t+1)$ are determined according to (18), (19) and [107]:

$$x_i(t+1) = x_i(t) + v_i(t+1) \tag{18}$$

$$v_i(t+1) = \omega v_1(t) + r_1 a_1 (P_{iBest}(t) - x_i(t)) + r_2 a_2 (G_{Best}(t) - x_i(t)) \tag{19}$$

where $\omega$ is the inertia weight; $v_i(t)$ the velocity at a time step *t*; $r_1$, $r_2$ are the randomly generated and uniformly distributed numbers ($0 \le r_1 \le 1$ and $0 \le r_2 \le 1$), respectively; $a_1$, $a_2$ are the acceleration coefficients; $P_{iBest}$ is the personal best; and $G_{Best}$ is the global best.

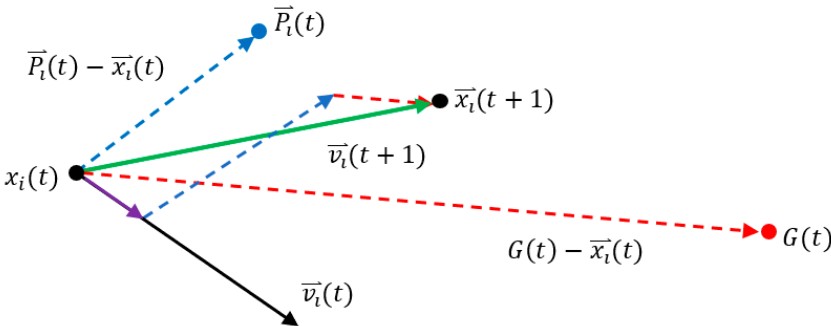

**Figure 5.** Geometric representation of the PSO model.

In (18) and (19), the heuristic parameters critically affect the optimization mechanism. The inertia weight $\omega$ critically affects the convergence behavior of the PSO mechanism. This fast convergence behavior occurs in the PSO mechanism because the inertia weight $\omega$ highly influences the exploration process of the PSO mechanism. This takes place due to the inertia weight creating an impact on the current velocity based on the previous velocity of each particle. Therefore, a large $\omega$ value toward the particles moves over the hyperspace rapidly, and the smaller $\omega$ value towards particles moves around its neighborhood particles. In this case, the large $\omega$ values allow global exploration, and smaller $\omega$ values create a tendency for exploitation or local exploration (a fine-tuning process).

Equation (19) shows that in the original PSO mechanism [107], the properly tuned values of parameters $a_1$ and $a_2$ allow for rapid convergence. The extended practical experiments show that this rapid convergence occurs with a large value of the cognitive parameter $a_1$ and with a large value of the social parameter $a_2$ when $a_1 + a_2 \le 4$. [108]. The values $r_1$ and $r_2$ control the diversity of the optimal solutions, and originally they are uniformly distributed between the range of 0 and 1.

However, it has been noticed that during the programme execution process, once the best particle in the global range traps in a local minimum, all particles follow that particle and are trapped in the same local minimum. In such cases, once they are trapped in the local minima, to overcome this issue the parameter values in (19) are switched via a separately dedicated Takagi–Sugeno–Kang (TSK) FLC. The design and development of the TSK-FLC have been discussed in detail in Section 7. Finally, this mechanism enables the particles to find a completely new solution set for the next generation.

Figures 1 and 2 show that the proposed metaheuristic FLC (TSK-PSO-FLC) has been designed and developed according to the closed-loop control strategies. Therefore, the proposed metaheuristic FLC is a dynamic controller that mimics the error $E$ and the rate of change of the error $dE$ and rapidly optimizes its own parameters for every sampling instant "$k$".

Therefore, this proposed Mamdani-approach-based metaheuristic FLC controller consists of a dynamic fuzzy model and a predefined static fuzzy model (with predefined fuzzy sets and fuzzy inference), as described in Section 3. The static fuzzy model consists of twenty-one (21) linguistic variables *(shown in* Figure 4*, seven (7) linguistic variables for each fuzzy set E, dE and U)*, twenty-one (21) membership functions *(seven (7) memberships functions for each fuzzy set E, dE and U)* and forty-nine (49 (=$7^2$)) fuzzy rule base functions.

## 6. Implementation of the Dynamic Fuzzy Logic Controller (Dynamic FLC)

In addition to this static fuzzy model, the dynamic behavior for this metaheuristic FLC is given through the dynamic PSO mechanism, whereby the PSO parameters are optimized dynamically via the predefined TSK-FLC. Figure 2 illustrates that all three main controllers are combined and synchronized together to act on the same sample instant "$k$".

Due to the non-linear parameter variations (uncertainties) of the electric rover, the proposed metaheuristic FLC should have the ability to quickly optimize the fuzzification process, de-fuzzification process and fuzzy reasoning process.

*6.1. Dynamic Fuzzification Process for the Fuzzy Antecedent and Consequent Dimensions with the Dynamic PSO Mechanism*

Figure 6 shows that when designing and developing the fuzzy antecedent and consequent dimensions for the proposed metaheuristic FLC, interval-valued fuzzy sets were taken into consideration. In these types of fuzzy sets, for any given input $x$, the membership in this fuzzy set $\widetilde{A}$ can be expressed as $\mu_A(x)$ for the membership interval from $\lambda_1$ to $\lambda_2$. To obtain more robust behavior for these types of fuzzy sets, the fuzzy intervals become fuzzy. These types of fuzzy functions can be represented by the "$n$" number of ordinary fuzzy sets. Therefore, these are called interval-valued fuzzy sets or type-2 fuzzy sets. These fuzzy sets can be expressed as (20):

$$\widetilde{A} : X \rightarrow \varepsilon([0,1]) \tag{20}$$

In this fuzzy set $\widetilde{A}$, $\varepsilon([0,1])$ includes all of the closed intervals of real numbers in [0,1]. Therefore, this can be represented as (21):

$$\varepsilon([0,1]) \subset \Re \tag{21}$$

In this case, Figure 6 shows that for each value of $x$, in $\widetilde{A}(x)$ the membership grade is given by the shaded area enclosed by the membership curves $f_1$ and $f_2$ When $x = a$, $f_1 \rightarrow \lambda_1$ and $f_2 \rightarrow \lambda_2$. Therefore, $\lambda_1$ and $\lambda_2$ are the upper bound limit and the lower bound limit for $\widetilde{A}(a)$ when $x = a$. In fuzzy-logic-based systems, the accuracy is critically dependent on the capabilities of designing and constructing the appropriate membership functions. However, reasonable values between the upper bound and the lower bound limits are selected. That means that the uncertainty behavior is not taken into consideration. In practice, this causes more speed fluctuations, due to the rapid changes that take place in the desired input trajectory. As per the observed data, interval-valued fuzzy linguistic

variables are more capable of capturing the uncertainty behavior of $E_k$ and $dE_k$. This is a more accurate process compared to using crisp variables. However, the disadvantage of these interval-valued membership functions compared to the ordinary fuzzy sets is that they are computationally more demanding and sophisticated.

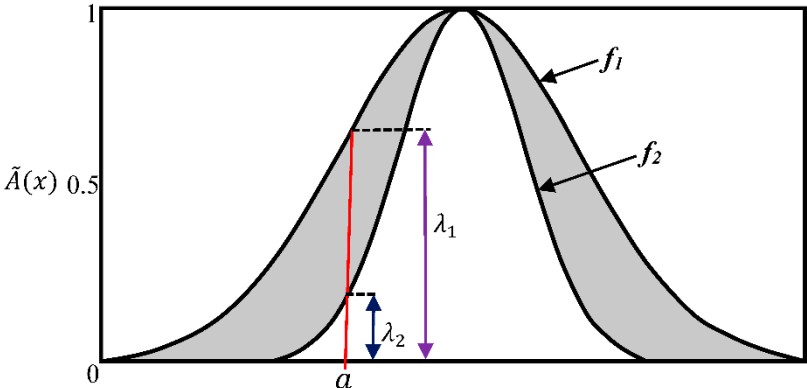

**Figure 6.** An interval-valued fuzzy set $\left( \tilde{A}(a) = [\lambda_1, \lambda_2] \right)$.

Practically, programming libraries and toolboxes are not available. This is because most modern day-to-day fuzzy-logic-based applications are not very sensitive to minor changes in the predefined fuzzy membership functions. However, the developed electric rover should respond to minor changes in the controlled parameters affecting the power optimization and stability of the rover.

Figure 6 shows that this interval-valued fuzzy set could be represented by two ordinary symmetric static Gaussian curve membership functions. In that case, the boundary limit functions of this interval-valued fuzzy membership function could be expressed as in (22) and (23):

$$f_1(x; \sigma_1; c_1) = \exp\left( -\frac{1}{2} \left( \frac{x - c_1}{\sigma_1} \right)^2 \right) \tag{22}$$

$$f_2(x; \sigma_2; c_2) = \exp\left( -\frac{1}{2} \left( \frac{x - c_2}{\sigma_2} \right)^2 \right) \tag{23}$$

In this curve, where $x$ represents the error $E_k$ and the change of the error $dE_k$ for any current situation, $\sigma_1$ and $\sigma_2$ represent the spread or width of the curves according to the maximum r.p.m. of the motor in the clockwise and counter-clockwise directions ($-5000$ r.p.m. to $5000$ r.p.m.). Here, $c_1$ and $c_2$ represent the position of the center of each membership curve during the fuzzification process.

According to (22) and (23), and as shown in Figure 6, the positions of the center $c$ and the spread or the width $\sigma$ of each the boundary curves $f_1$ and $f_2$ are fixed values. However, according to the non-linear behavior of this electric rover, the fuzzy membership functions should be more robust. Therefore, these interval-valued membership functions limit the optimum fuzzification process and the optimum fuzzy reasoning process because of the fixed boundaries.

To overcome these drawbacks while maintaining the properties of the interval-valued membership functions, a dynamic fuzzification process was designed and developed. The dynamic behavior and the robust performance are given to the Mamdani-based dynamic FLC during the fuzzification, defuzzification and fuzzy reasoning processes through a dynamically optimized metaheuristic PSO mechanism for every sampling instant "$k$". The proposed dynamic PSO mechanism tunes 22 parameters of the proposed dynamic meta-heuristic FLC. These 22 parameters of the dynamic FLC belong to the dynamic membership functions in the fuzzification process, de-fuzzification process and dynamic fuzzy reasoning process (the dynamic fuzzy rule base).

Therefore, in this proposed dynamic metaheuristic FLC, the membership functions are formed with dynamic Gaussian curve membership functions.

Figure 7 and (24) show the dynamic optimization mechanism of each fuzzy linguistic variable or a fuzzy membership function for any given sampling instant "$k$".

$$f_i(x; \sigma_i; c_i) = \exp\left( -\frac{1}{2}\left(\frac{x - c_i}{\sigma_i}\right)^2 \right) \tag{24}$$

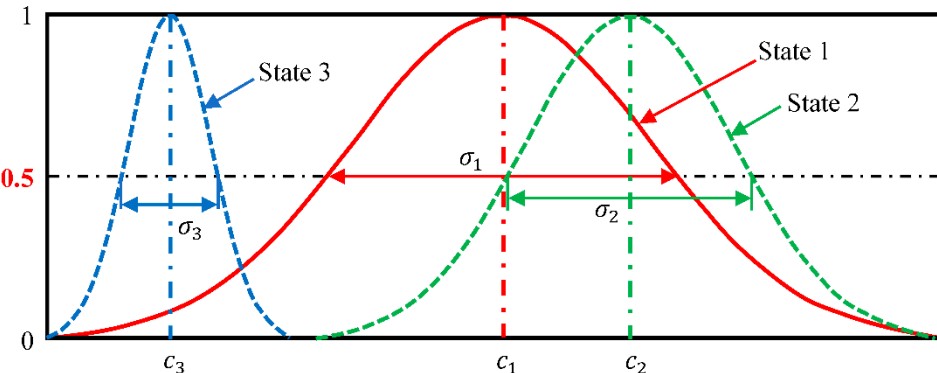

**Figure 7.** Dynamic behavior of the positive medium $PM(E_k)$ linguistic variable.

According to (24), to dynamically optimize the values $c_i$ and $\sigma_i$ of each membership function, two dynamic *membership gain factors* $gain_{c_i}$ and $gain_{\sigma_i}$ are considered. Therefore, for all seven linguistic variables ($NB$, $NM$, $NS$, $ZE$, $PS$, $PM$ and $PB$), 14 dynamic membership gain factors are taken into consideration. Therefore, the peak value of the center $c_i$ and the width or the standard deviation $\sigma_i$ of each membership function are functions of these dynamic membership gain factors.

In this proposed dynamic metaheuristic FLC, $E_k$ and $dE_k$ are the inputs to the controller, where $U_k$ is the output (actual r.p.m.). For all input and output variables, 21 ($7 \times 3$) linguistic variables take place (each input or output variable is formed with seven linguistic variables). However, as described above, according to this mechanism, to dynamically optimize all these 21 linguistic variables that belong to $E_k$, $dE_k$ and $U_k$, 42 ($14 \times 3$) optimized membership gain factors ($gain_{c_i}$ and $gain_{\sigma_i}$) need to be generated through the proposed dynamic PSO mechanism.

However, these 42 membership gain factors will be good enough only to optimize the angular velocity of one wheel.

Therefore, when considering the non-linear behavior of this four-wheeled independent-drive electric rover to optimize the desired angular velocity of all four wheels, 168 ($42 \times 4$) dynamic membership gain factors need to be generated and optimized. Therefore, in the hyperspace of the PSO mechanism, 168 dimensions need to be considered for each particle when it moves to a global best position $G_{Best}$ while optimizing each particle to its personal best position $P_{Best}$.

This makes the proposed controller more sophisticated and computationally inefficient. In such cases, to dynamically optimize all 21 linguistic variables that belong to the input–output variables ($E_k$, $dE_k$ and $U_k$), another three independent gain factors were taken into consideration. These input–output (IO) gain factors $\lambda_E$, $\lambda_{dE}$ and $\lambda_U$ belong to the input–output variables $E_k$, $dE_k$ and $U_k$, respectively. In the PSO mechanism, $\lambda_E$, $\lambda_{dE}$ and $\lambda_U$ are optimized on another independent three-dimensional axis.

Therefore, when considering the optimizing process of one wheel, to enhance the dynamic performance of the PSO mechanism and to simplify the algorithm, out of these 42 membership gain factors (for the 21 linguistic variables) only 14 membership gain factors ($gain_{c_i}$ and $gain_{\sigma_i}$) are optimized. However, these 14 membership gain factors are good enough only to optimize seven linguistic variables. In this case, as a solution, these membership gain factors are optimized through consequent 14 dimensions of the

PSO mechanism by considering the objective function index threshold level (a function of the $E_k$).

These fourteen (14) common membership gain factors are utilized to generate the optimized $c_i$ and $\sigma_i$ values for all 21 linguistic variables by taking into consideration the optimized IO gains factors ($\lambda_E$, $\lambda_{dE}$ and $\lambda_U$). The dynamically optimized *14 common membership gain factors* through the global best $G_{Best}$ solution of the PSO mechanism can be expressed as shown in Table 3.

**Table 3.** The fourteen (14) common membership gain factors optimized by the PSO mechanism.

| $C_i$ (Optimized Peak Value or the Center Value) | j | $\sigma_i$ (Optimized Standard Deviation) | j |
|---|---|---|---|
| $gain_{c_1} = -\frac{1}{\alpha_1} G_{(Best)_j} \times 5000$ | 1 | $gain_{\sigma_1} = \frac{1}{\beta_1} G_{(Best)_j} \times 707$ | 2 |
| $gain_{c_2} = -\frac{1}{\alpha_2} G_{(Best)_j} \times 3333$ | 3 | $gain_{\sigma_2} = \frac{1}{\beta_2} G_{(Best)_j} \times 707$ | 4 |
| $gain_{c_3} = -\frac{1}{\alpha_3} G_{(Best)_j} \times 1667$ | 5 | $gain_{\sigma_3} = \frac{1}{\beta_3} G_{(Best)_j} \times 707$ | 6 |
| $gain_{c_4} = \pm \frac{1}{\alpha_4} G_{(Best)_j}$ | 7 | $gain_{\sigma_4} = \frac{1}{\beta_4} G_{(Best)_j} \times 707$ | 8 |
| $gain_{c_5} = \frac{1}{\alpha_5} G_{(Best)_j} \times 1667$ | 9 | $gain_{\sigma_5} = \frac{1}{\beta_5} G_{(Best)_j} \times 707$ | 10 |
| $gain_{c_6} = \frac{1}{\alpha_6} G_{(Best)_j} \times 3333$ | 11 | $gain_{\sigma_6} = \frac{G}{\beta_6} G_{(Best)_j} \times 707$ | 12 |
| $gain_{c_7} = \frac{1}{\alpha_7} G_{(Best)_j} \times 5000$ | 13 | $gain_{\sigma_7} = \frac{G}{\beta_7} G_{(Best)_j} \times 707$ | 14 |

Where, $\alpha_i$, $\beta_i \in \mathbb{R}$, $1 \leq i \leq 7$, $i \in \mathbb{Z}^+$, $1 \leq j \leq 14$, $j \in \mathbb{Z}^+$ and $G_{(Best)_j}$ represent the fourteen (14)-dimensional global best solution of the PSO mechanism in Table 3.

Based on these common membership gain factors and the IO gain factors, the optimized $c_i$ and $\sigma_i$ values for all 21 linguistic variables can be expressed, as shown in Tables 4–6. Here, $\lambda_E$ is the I/O gain factor of the $E_k$ membership functions $p_{E_i}$, $q_{E_i} \in \mathbb{R}$, $1 \leq i \leq 7$ and $i \in \mathbb{Z}^+$.

Here, $\lambda_{dE}$ is the I/O gain factor of the $dE_k$ membership functions $p_{dE_i}$, $q_{dE_i} \in \mathbb{R}$, $1 \leq i \leq 7$ and $i \in \mathbb{Z}^+$ in Table 5.

Here, $\lambda_U$ is the I/O gain factor of the $U_k$ membership functions $p_{U_i}$, $q_{U_i} \in \mathbb{R}$, $1 \leq i \leq 7$ and $i \in \mathbb{Z}^+$ shown in Table 6.

This PSO-based dynamic FLC optimization mechanism is used to optimize the angular velocity of one wheel in a highly non-linear environment.

**Table 4.** The dynamically optimized $c_i$ and $\sigma_i$ factors for the membership functions of $E_k$.

| Optimized Gain Factors for the Input Variable $E_k$ (Error) | | |
|---|---|---|
| $c_i$ (Optimized Peak Value) | $\sigma_i$ (Optimized Standard Deviation or the Width of the Curve) | Membership Function (MF) |
| $c_{E_1} = -5000 \pm \left(\frac{\lambda_E}{p_{E_1}}\right) \times gain_{c_1}$ | $\sigma_{E_1} = 707 \pm \left(\frac{\lambda_E}{q_{E_1}}\right) \times gain_{\sigma_1}$ | NB |
| $c_{E_2} = -3333 \pm \left(\frac{\lambda_E}{p_{E_2}}\right) \times gain_{c_2}$ | $\sigma_{E_2} = 707 \pm \left(\frac{\lambda_E}{q_{E_2}}\right) \times gain_{\sigma_2}$ | NM |
| $c_{E_3} = -1667 \pm \left(\frac{\lambda_E}{p_{E_3}}\right) \times gain_{c_3}$ | $\sigma_{E_3} = 707 \pm \left(\frac{\lambda_E}{q_{E_3}}\right) \times gain_{\sigma_3}$ | NS |
| $c_{E_4} = \left(\frac{\lambda_E}{p_{E_4}}\right) \times gain_{c_4}$ | $\sigma_{E_4} = 707 \pm \left(\frac{\lambda_E}{q_{E_4}}\right) \times gain_{\sigma_4}$ | Z |
| $c_{E_5} = 1667 \pm \left(\frac{\lambda_E}{p_{E_5}}\right) \times gain_{c_5}$ | $\sigma_{E_5} = 707 \pm \left(\frac{\lambda_E}{q_{E_5}}\right) \times gain_{\sigma_5}$ | PS |
| $c_{E_6} = 3333 \pm \left(\frac{\lambda_E}{p_{E_6}}\right) \times gain_{c_6}$ | $\sigma_{E_6} = 707 \pm \left(\frac{\lambda_E}{q_{E_6}}\right) \times gain_{\sigma_6}$ | PM |
| $c_{E_7} = 5000 \pm \left(\frac{\lambda_E}{p_{E_7}}\right) \times gain_{c_7}$ | $\sigma_{E_7} = 707 \pm \left(\frac{\lambda_E}{q_{E_7}}\right) \times gain_{\sigma_7}$ | PB |

**Table 5.** The dynamically optimized $c_i$ and $\sigma_i$ factors for the membership functions of $dE_k$.

| $c_i$ (Optimized Peak Value) | $\sigma_i$ (Optimized Standard Deviation or the Width of the Curve) | Membership Function (MF) |
|---|---|---|
| | **Optimized Gain Factors for the Input Variable $dE_k$ (Error)** | |
| $c_{dE_1} = -5000 \pm \left(\frac{\lambda_{dE}}{p_{dE_1}}\right) \times gain_{c_1}$ | $\sigma_{dE_1} = 707 \pm \left(\frac{\lambda_{dE}}{q_{dE_1}}\right) \times gain_{\sigma_1}$ | NB |
| $c_{dE_2} = -3333 \pm \left(\frac{\lambda_{dE}}{p_{dE_2}}\right) \times gain_{c_2}$ | $\sigma_{dE_2} = 707 \pm \left(\frac{\lambda_{dE}}{q_{dE_2}}\right) \times gain_{\sigma_2}$ | NM |
| $c_{dE_3} = -1667 \pm \left(\frac{\lambda_{dE}}{p_{dE_3}}\right) \times gain_{c_3}$ | $\sigma_{dE_3} = 707 \pm \left(\frac{\lambda_{dE}}{q_{dE_3}}\right) \times gain_{\sigma_3}$ | NS |
| $c_{dE_4} = \left(\frac{\lambda_{dE}}{p_{dE_4}}\right) \times gain_{c_4}$ | $\sigma_{dE_4} = 707 \pm \left(\frac{\lambda_{dE}}{q_{dE_4}}\right) \times gain_{\sigma_4}$ | Z |
| $c_{dE_5} = 1667 \pm \left(\frac{\lambda_{dE}}{p_{dE_5}}\right) \times gain_{c_5}$ | $\sigma_{dE_5} = 707 \pm \left(\frac{\lambda_{dE}}{q_{dE_5}}\right) \times gain_{\sigma_5}$ | PS |
| $c_{dE_6} = 3333 \pm \left(\frac{\lambda_{dE}}{p_{dE_6}}\right) \times gain_{c_6}$ | $\sigma_{dE_6} = 707 \pm \left(\frac{\lambda_{dE}}{q_{dE_6}}\right) \times gain_{\sigma_6}$ | PM |
| $c_{dE_7} = 5000 \pm \left(\frac{\lambda_{dE}}{p_{dE_7}}\right) \times gain_{c_7}$ | $\sigma_{dE_7} = 707 \pm \left(\frac{\lambda_{dE}}{q_{dE_7}}\right) \times gain_{\sigma_7}$ | PB |

**Table 6.** The dynamically optimized $c_i$ and $\sigma_i$ factors for the membership functions of $U_k$.

| $c_i$ (Optimized Peak Value) | $\sigma_i$ (Optimized Standard Deviation or the Width of the Curve) | Membership Function (MF) |
|---|---|---|
| | **Optimized Gain Factors for the Input Variable $U_k$ (Error)** | |
| $c_{U_1} = -5000 \pm \left(\frac{\lambda_U}{p_{u_1}}\right) \times gain_{c_1}$ | $\sigma_{U_1} = 707 \pm \left(\frac{\lambda_U}{q_{u_1}}\right) \times gain_{\sigma_1}$ | NB |
| $c_{U_2} = -3333 \pm \left(\frac{\lambda_U}{p_{u_2}}\right) \times gain_{c_2}$ | $\sigma_{U_2} = 707 \pm \left(\frac{\lambda_U}{q_{u_2}}\right) \times gain_{\sigma_2}$ | NM |
| $c_{U_3} = -1667 \pm \left(\frac{\lambda_U}{p_{u_3}}\right) \times gain_{c_3}$ | $\sigma_{U_3} = 707 \pm \left(\frac{\lambda_U}{q_{u_3}}\right) \times gain_{\sigma_3}$ | NS |
| $c_{U_4} = \left(\frac{\lambda_U}{p_{u_4}}\right) \times gain_{c_4}$ | $\sigma_{U_4} = 707 \pm \left(\frac{\lambda_U}{q_{u_4}}\right) \times gain_{\sigma_4}$ | Z |
| $c_{U_5} = 1667 \pm \left(\frac{\lambda_U}{p_{u_5}}\right) \times gain_{c_5}$ | $\sigma_{U_5} = 707 \pm \left(\frac{\lambda_U}{q_{u_5}}\right) \times gain_{\sigma_5}$ | PS |
| $c_{U_6} = 3333 \pm \left(\frac{\lambda_U}{p_{u_6}}\right) \times gain_{c_6}$ | $\sigma_{U_6} = 707 \pm \left(\frac{\lambda_U}{q_{u_6}}\right) \times gain_{\sigma_6}$ | PM |
| $c_{U_7} = 5000 \pm \left(\frac{\lambda_U}{p_{u_7}}\right) \times gain_{c_7}$ | $\sigma_{U_7} = 707 \pm \left(\frac{\lambda_U}{q_{u_7}}\right) \times gain_{\sigma_7}$ | PB |

For example, Figure 7 shows that out of these seven linguistic variables, if the positive medium (PM) linguistic variable (PM membership function) is considered, the optimized state 1, state 2 and state 3 of the PM membership function can be obtained for the sampling instants "$k$", "$k + 1$" and "$k + 2$", respectively. When the membership function shape changes dynamically through this optimization mechanism, this causes the membership grade $\mu_A(x)$ to become more dynamic for every sample instant $k$. This enables one to capture the uncertainty behavior as the interval-valued membership function. Figure 8 shows the overall mechanism of the dynamic fuzzification process through the dynamic PSO mechanism.

Figure A1 shows the optimized $c_i$ and $\sigma_i$ values obtained for all 21 linguistic variables. Moreover, according to Figure A1, this dynamic membership function optimization mechanism is mainly based on the dynamically optimized 14 common membership gain factors and the 3 dynamic I/O gain factors.

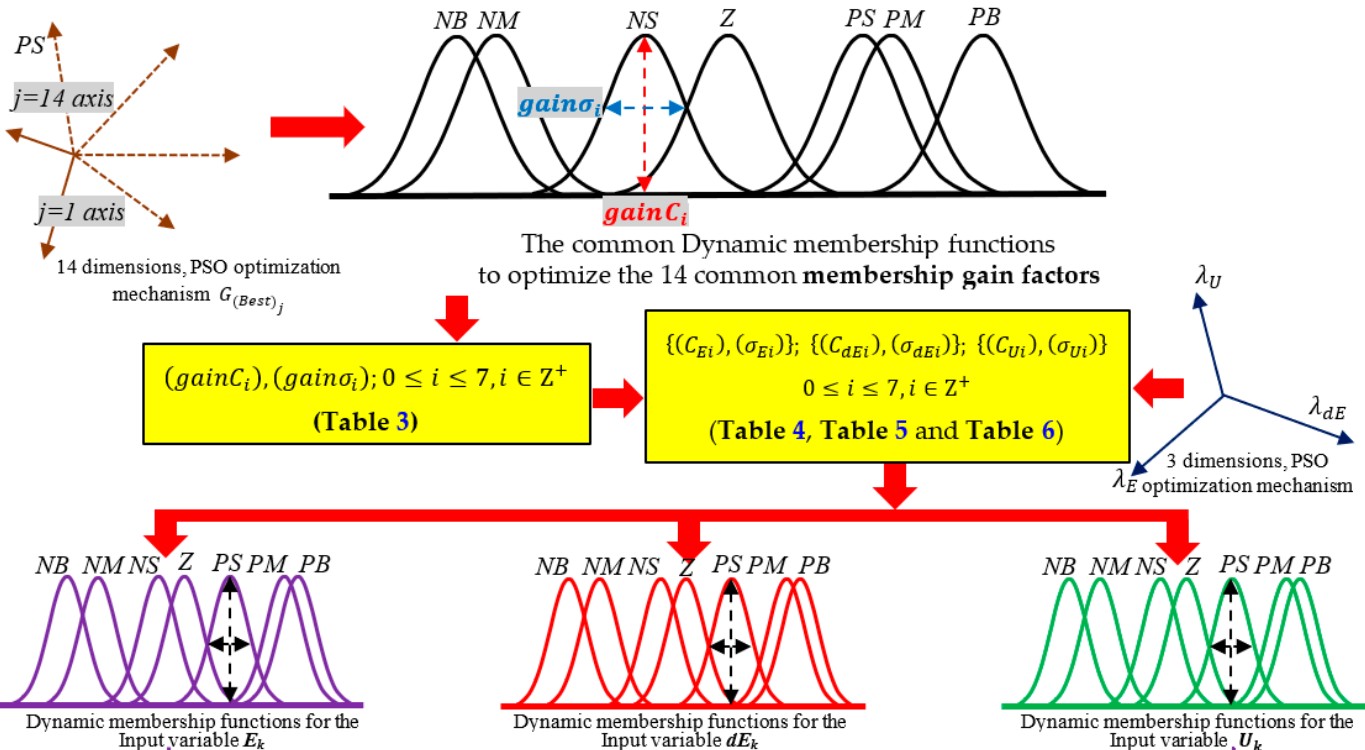

**Figure 8.** The overall mechanism of the dynamic fuzzification process through the dynamic PSO mechanism.

*6.2. Dynamic Fuzzy Reasoning Process Optimized by the PSO*

Figure 8 shows that in addition to the 17 dimensions of the dynamic fuzzification process, another additional 5 consequent dimensions were taken into consideration. The first 3 consequent axes represent the linguistic variables, and in the other remaining two axes one axis represents either the "and" fuzzy operator (minimum) or the "or" fuzzy operator (maximum) between the consequent and the other remaining axis, which represents the weight of the fuzzy rule, as shown in Figure 9. In the first three axes (axis: $j = 18$, 19 and 20) the lower bound value $L_{Bj}$ and the upper bound value $U_{Bj}$ were limited to 1 and 7, respectively.

However, in practice, it was noted during the optimization process that when the particles moved to the global best position $G_{(Best)_j}$ they generated floating values. Therefore, the generated optimized values of axes 18, 19 and 20 were rounded and assigned to the most suitable linguistic variables of the inputs and output fuzzy variables. In this dynamic metaheuristic FLC, the corresponding global solution values of each dimension are "$NB = 1$", "$NM = 2$", "$NS = 3$", "$Z = 4$", "$PS = 5$", "$PM = 6$" and "$PB = 7$". The lower bound limit and the upper bound limit of the other remaining axes (axis 21 and axis 22) are limited to 0 and 1, respectively. However, on axis 21, the obtained optimized values are rounded. If the rounded value is 0 or 1 then it is assigned to the logical operator "or" or "and", respectively. The 22-axis value (value between 0 and 1), which is generated through the PSO mechanism, is used as the firing strength of the fuzzy rule.

All of these 5-dimensional optimized values are obtained over and over again for every sampling instant "$k$" to generate the appropriate dynamic fuzzy rules in the fuzzy reasoning process.

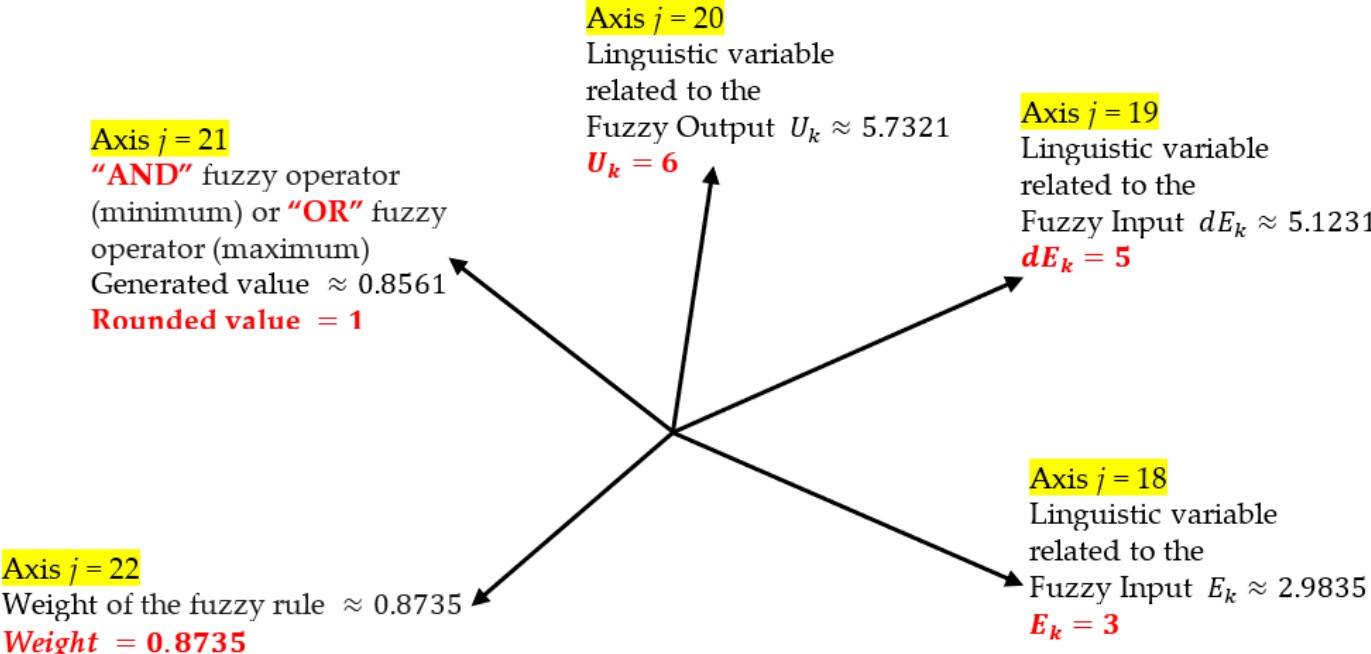

**Figure 9.** The five consequent dimensions of the PSO mechanism for the dynamic fuzzy reasoning process.

Table 7 shows all corresponding decimal values that are assigned to each linguistic variable of each membership function. Therefore, Figure 9 shows the dynamically generated fuzzy rule through the dynamic PSO mechanism in the fuzzy reasoning process, which can be expressed as in Table 8.

**Table 7.** Corresponding decimal values for the linguistic variables in the PSO mechanism.

| | Linguistic Variables | | | | | | |
|---|---|---|---|---|---|---|---|
| | **NB** | **NM** | **NS** | **ZE** | **PS** | **PM** | **PB** |
| Rounded decimal value in Axis **18**, **19** and **20**. | 1 | 2 | 3 | 4 | 5 | 6 | 7 |

**Table 8.** Dynamically generated fuzzy rules through the dynamic PSO mechanism.

| Fuzzy variables | : | $E_k$ | $dE_k$ | $U_k$ |
|---|---|---|---|---|
| Decimal representation | : | 3 | 5 | 6 |
| Corresponding fuzzy rule | : | If the $E_k$ is **NS** and the $dE_k$ is **PS** then $U_k$ is **PM** | | |

Here, $U_k$ is the output angular speed should be of the permanent magnet BLDC motor.

In addition to these dynamically generated fuzzy rules through the PSO mechanism, 49 static fuzzy rules are also taken into consideration when optimizing the behavior of the BLDC motor as illustrated in Figure 2 and Table 2. Figure 10 shows the designed and developed program structure used to integrate the static fuzzy rule mechanism and the dynamic fuzzy rule mechanism into the proposed metaheuristic FLC. In the optimization mechanism, these static and dynamic fuzzy reasoning processes are synchronized for every sampling instant "*k*" However, Figure 2 shows that the proposed robust PSO mechanism was optimized through the TSK-FLC.

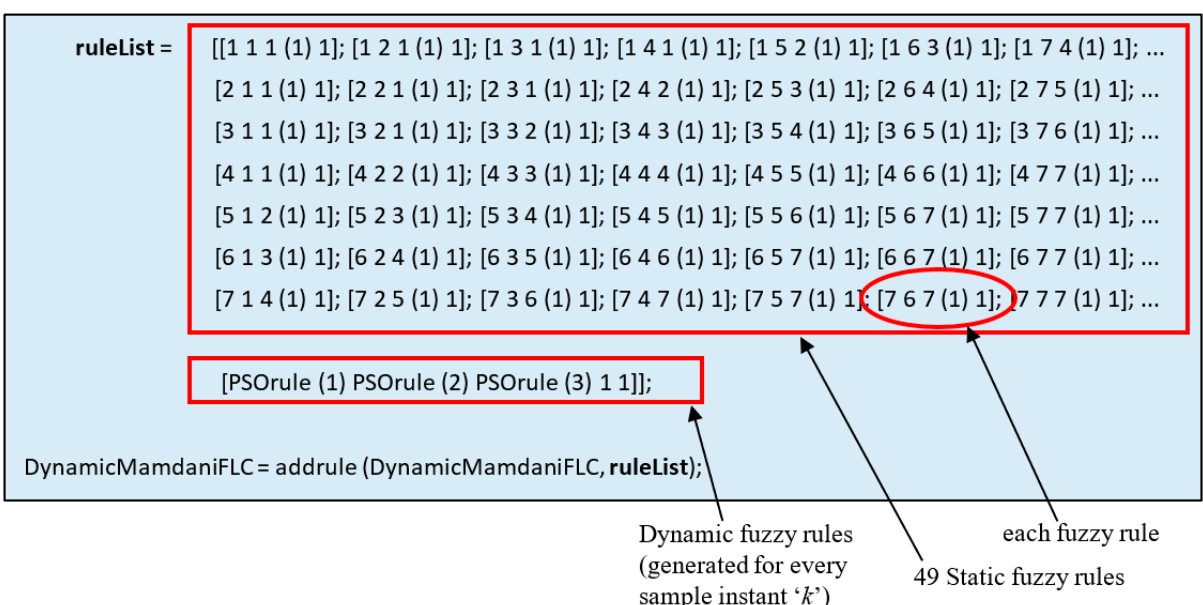

**Figure 10.** The PSO program structure of the metaheuristic fuzzy rule base.

*6.3. Optimization of the Dynamic PSO Mechanism to Tune the Proposed Metaherustic FLC*

The tuning process of the proposed metaheuristic FLC utilizing 22 variables is described in Section 6.2. In order to dynamically tune these 22 variables, 22 dimensions are taken into consideration in the hyperspace when designing the proposed dynamic PSO mechanism.

In a traditional PSO mechanism [107], all candidate solutions or particles are unable to converge to an equilibrium position due to the rapid changes in the amplitude and the frequency of the desired input trajectory (the dynamic behavior) of each wheel. To overcome this problem, in the proposed metaheuristic FLC, the parameters of the dynamic PSO mechanism are tuned in real-time through a separately dedicated TSK-FLC. Through this TSK-FLC, for every iteration or discrete sample, the particle positions (solutions) are re-initialized by optimizing the population size $nPop$, inertia weights ($\omega_{max}$ and $\omega_{min}$) and acceleration coefficients ($a_1$ and $a_2$) and the PSO is charged by adding dynamic acceleration $a_{ij}$ into Equation (19). Finally, this 22-dimensional dynamic PSO mechanism can be expressed as shown in (25) and (26):

$$x_{ij}(t+1) = x_{ij}(t) + v_{ij}(t+1) \tag{25}$$

$$v_{ij}(t+1) = \omega v_{ij}(t) + r_1 a_1 \left( P_{(Best)_{ij}}(t) - x_{ij}(t) \right) + r_2 a_2 \left( G_{(Best)_j}(t) - x_{ij}(t) \right) + a_{ij}(t). \tag{26}$$

where $a_{ij}(t) = \frac{1}{K} \left[ \frac{\left( \sum_{i=1;\, p=1}^{i=\infty;\, p=nPop} a_{ip} \right)}{(nPop)} \right]$, $K \in \mathbb{R}$, $nPop$ is the population size, $j \in \mathbb{Z}^+$, $1 \le j \le 18$, $r_1 a_1 \left( P_{(Best)_{ij}}(t) - x_{ij}(t) \right)$ is the cognitive component and $r_2 a_2 \left( G_{(Best)_j}(t) - x_{ij}(t) \right)$ is the social component.

The cognitive component and the social component are taken into consideration for the previous best solution and the global best solution, respectively.

In this mathematical model, for the proposed dynamic metaheuristic FLC, the inertia weight $\omega$ is computed according to (27) and continuously optimized through the TSK-FLC as described in Section 7.

$$\omega = \omega_{max} - \left( \frac{\omega_{max} - \omega_{min}}{maxIterations} \right) \times Iteration \tag{27}$$

where $\omega_{max}$ and $\omega_{min}$ are the maximum and minimum inertia weights assigned in the TSK-FLC, "*maxIterations*" is the maximum number of iterations and "*Iteration*" is the current iteration.

To optimize the metaheuristic FLC via the dynamic PSO mechanism, an objective function $f_{ob}$ is utilized as the input to the PSO mechanism, which is a function of the $E_k$ and the $dE_k$, as shown in Figure 2. In this dynamic environment, for rapid convergence and enhancement of the performance of the PSO mechanism, an objective function index $I_{(f_{ob})}$ is taken into consideration while applying a penalty on each constraint violation. Here, $I_{(f_{ob})}$ is computed according to (28) and (29):

$$f_{ob} = RMSE = \sqrt{\frac{1}{N} \sum_{k=1}^{N} (E_k(t))^2} \tag{28}$$

$$I_{(f_{ob})} = f_{ob} + \left( \sum_{k=1}^{N} E_k \right) \times \text{penalty} \tag{29}$$

where $\sqrt{\frac{1}{N} \sum_{k=1}^{N} (E_k(t))^2}$ is the root mean square error.

## 7. Implementation of the Takagi–Sugeno–Kang (TSK) FLC

The output level "Z" of the TSK-FLC can be expressed as (30):

$$Z_k = a(E_k) + b(dE_k) + c. \tag{30}$$

The constructed TSK-FLC contains 16 fuzzy rules, where $a$, $b$ and $c$ are constants and ($a = b = 0$). This means that the output level is always a constant and not a linear function. When the firing strength of a particular fuzzy rule is $w_k$ for a $k$th sample and when each fuzzy rule is interacting with an "and" operator, then the firing strength can be defined as (31):

$$w_k = Andmethod(MF_i(E), MF_i(dE)). \tag{31}$$

where $MF_i(E)$ and $MF_i(dE)$ are input membership functions for the $i$th linguistic variables ($i \in \mathbb{Z}^+$ and $1 \leq i \leq 7$). In the TSK-FLC, for all the aforementioned 16 fuzzy rules, the weighted average final output is computed as (32):

$$Output_{(TSK-FL)}\text{i} = \frac{\sum_{i=1}^{16} w_i Z_i}{\sum_{i=1}^{16} w_i} \tag{32}$$

The output of the TSK-FLC is determined according to (30)–(32). Figure 11 shows the 6 output parameters of the TSK-FLC, which are inputs to the dynamic PSO mechanism and tuned in real-time according to the rapid changes of the $E_k$ and $dE_k$.

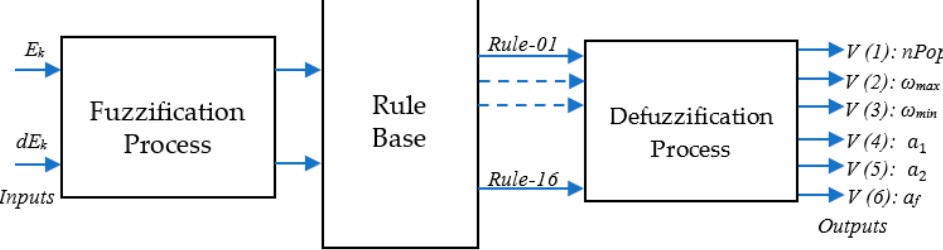

**Figure 11.** Block diagram of the TSK-FLC.

Figure 12 shows the membership functions and boundaries of each fuzzy set (membership function), where "PVS", "PS", "PM" and "PB" represent the positive very small, positive small, positive medium and positive big linguistic variables, respectively. According to each output, the practically verified linguistic values of the linguistic variables are shown in Table 9. Table 10 shows the main governing fuzzy rules of the TSK-FLC. In the TSK-FLC, the outputs are always constants (not linear functions).

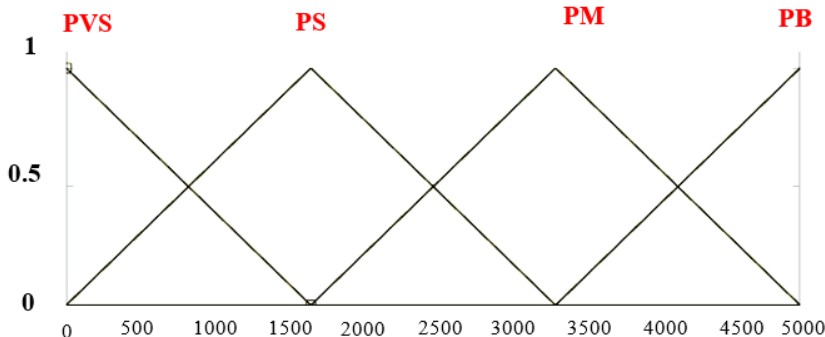

**Figure 12.** Membership functions of $E_k$ and $dE_k$.

**Table 9.** Assigned linguistic values for the linguistic variables of the TSK-FLC.

| | TSK-FL Controller Outputs | | | | | |
|---|---|---|---|---|---|---|
| Linguistic Variables | $nPop$ | $\omega_{max}$ | $\omega_{min}$ | $a_1$ | $a_2$ | $a_f$ |
| PVS | 5 | 0.800 | 0.100 | 0.975 | 0.975 | None |
| PS | 10 | 0.825 | 0.300 | 0.985 | 0.985 | 0.009 |
| PM | 15 | 0.875 | 0.400 | 0.995 | 0.995 | None |
| PB | 20 | 0.900 | 0.500 | 1.000 | 1.000 | None |

**Table 10.** Governing fuzzy rules of the proposed TSK-FLC.

| No. | Inputs | | | Outputs | | | | |
|---|---|---|---|---|---|---|---|---|
| | $E_k$ | $dE_k$ | $nPop$ | $\omega_{max}$ | $\omega_{min}$ | $a_1$ | $a_2$ | $a_f$ |
| 01 | PVS | PVS | PS | PM | PM | PM | PM | None |
| 02 | PS | PVS | PS | PM | PM | PM | PM | None |
| 03 | PM | PVS | PM | PB | PM | PM | PM | None |
| 04 | PB | PVS | PB | PB | PB | PB | PB | PS |
| 05 | PVS | PS | PS | PM | PM | PS | PS | None |
| 06 | PS | PS | PM | PM | PM | PM | PM | None |
| 07 | PM | PS | PM | PB | PB | PM | PM | None |
| 08 | PB | PS | PB | PB | PB | PB | PB | PS |
| 09 | PVS | PM | PM | PM | PM | PM | PM | None |
| 10 | PS | PM | PM | PB | PM | PM | PM | None |
| 11 | PM | PM | PM | PB | PB | PM | PM | None |
| 12 | PB | PM | PB | PB | PB | PB | PB | PS |
| 13 | PVS | PB | PM | PM | PM | PM | PM | None |
| 14 | PS | PB | PM | PB | PB | PM | PM | None |
| 15 | PM | PB | PB | PB | PB | PM | PM | None |
| 16 | PB | PB | PB | PB | PB | PB | PB | PS |

## 8. Modelling of Mechanical Dynamics of the Four-Wheeled Independent-Drive Electric Rover

The design and development of a dynamic metaheuristic FLC (TSK-PSO-FLC) to control a non-linear plate in the desired manner is described in Section 1. Therefore, the dynamic self-adaptive fuzzification process, defuzzification process and FIM are discussed instead of describing the electric rover system models. Figure 1 shows that the developed

intelligent dynamic controller of this four-wheeled independent-drive electric rover is a combination of 3 controllers.

A previous study [104] described the obtained rover system model that is used to identify the mechanical dynamics behavior of the rover, the proposed tire model that is utilized to identify the wheel slip of each tire, the tire model that is proposed to identify the kinetic friction coefficient ($\mu$) between the tire and the road surface and the designed and developed steering FLC and differential FLC.

## 9. Results and Discussion

Straight, incline, downhill and curving roads were used to examine the performance of the built dynamic metaheuristic FLC with all sorts of road limits during the quickest acceleration and deceleration phases. This article, on the other hand, focuses on road testing conducted on straight roads under various traction conditions (such as on slippery wet grass surfaces), as shown in Figures A2 and A3.

When the rover moves along a straight path (trajectory), even under varied traction environments (different friction coefficients $\mu_j$), the orientation of the rover should remain constant throughout time. Correspondingly, in order to maintain a certain intended orientation based on the observed data from the installed gyroscope, the built dynamic metaheuristic FLC should synchronize all of the angular velocities (r.p.m.) of each wheel in order to retain the desired orientation (fixed orientation) (because the developed rover turning method based on the differential speed technique is not discussed in this paper).

Figure 13 depicts the desired throttle signal and the desired steering angle, which indicate a zero-degree yaw angle (in its neutral position) during the straight road test operating time period (fed wirelessly based on the IEEE 802.15.04 protocol through a computer-controlled joystick).

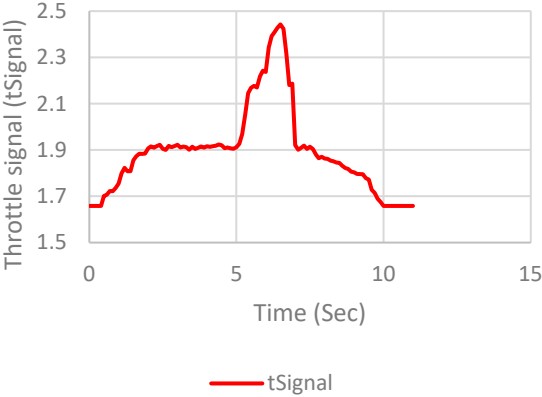

**Figure 13.** The throttle signal is given to the controller to achieve the desired translational velocity. The neutral position of the steering angle level is 1.7 (to achieve a zero-yaw angle) [104].

Figure 14 shows that the $\mu_j$ fluctuation over time, as calculated by Equation (7) in [104], is less than 0.4, while the average $\mu_j$ is about 0.12. This means that the wet grass road surface is extremely slick. Consequently, in all of these situations, optimizing the wheel slip of each wheel and maintaining the desired yaw angle across the operating time period proved very complex.

Figure 15 depicts the r.p.m. of each wheel in relation to the required average r.p.m *refWs*. Table 11 shows the steady-state error $E_{ss}$ and steady-state error as a percentage $E_{ss}\%$ for the peak edge of the r.p.m. of each independent wheel. The steady-state error $E_{ss}$ is minimized during the deceleration via the proposed FLC by driving the motor(s) in the opposite direction. However, this could be more effective by applying brake force during the deceleration via a fuzzy logic control magnetic brake mechanism [109,110] in addition to a reverse polarity changing mechanism. The greatest average r.p.m. and

translational velocity values of the rover are 1024.4 r.p.m. and 25.09 km/h, respectively, at the maximum r.p.m.

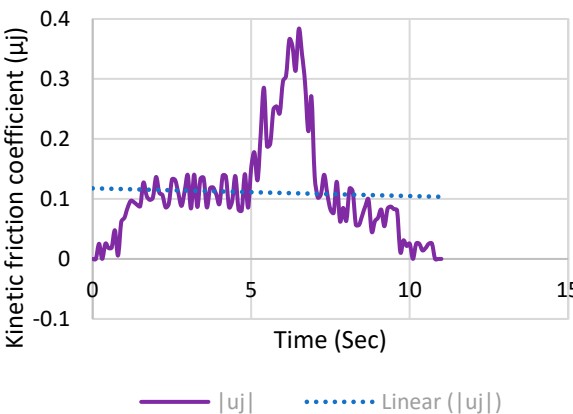

**Figure 14.** The variation of the kinetic friction coefficient ($\mu_j$) vs. time (Sec) [104].

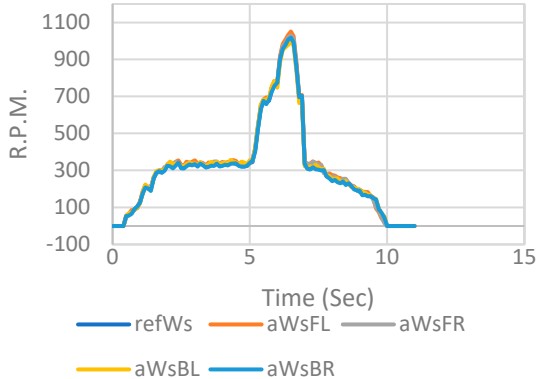

**Figure 15.** The reference r.p.m. (refWs) and the actual r.p.m. of each wheel vs. time [104].

**Table 11.** Steady-state error percentage (Ess%) of each independent wheel [104].

| Wheel of the Rover | Steady-State Error ($E_{ss}$) (r.p.m.) | Steady-State Error % ($E_{ss}$%) (r.p.m.) |
|---|---|---|
| Front-Left (FL) | 58.16 | 5.24 |
| Front-Right (FR) | 68.81 | 6.27 |
| Back-Left (BL) | −28.65 | −2.95 |
| Back-Right (BR) | −51.82 | −5.36 |

Figures 16 and 17 show the three orthogonal acceleration directions ("*AccX: longitudinal*", "*AccY: lateral*" and "*AccZ: radial*") and three gyro-angles ("*GyYaw: Yaw*", "*GyPit: Pitch*" and "*GyRol: Roll*") observed when the wheel speed data were examined.

Furthermore, Figure 16 demonstrates that as the rover achieved a high r.p.m., the radial acceleration varied dramatically. This occurred as a result of an uneven road surface.

Figure 17 shows that the rover encountered a slope of around 30 degrees within about 3.5 s. Again, within 3.5 s, the rover's actual yaw angle was pushed away from its target yaw angle by roughly 10 degrees. The yaw angle variation was 10 degrees in relation to the beginning location when compared to the overall travel distance (measured distance was 105.64 m). Furthermore, due to the slope of the grass surface into the lateral direction of the rover, a roll angle of roughly 2 degrees occurred.

Based on the observed wheel speed data, the wheel slip ratio, $S_j$, was estimated in real time for various road surfaces using Equation (3) in [104] (for different friction coefficients).

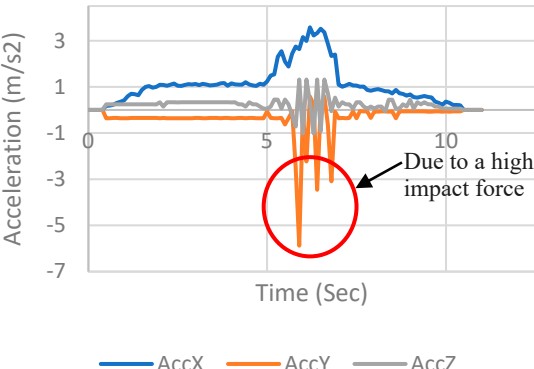

**Figure 16.** The actual three orthogonal acceleration directions vs. the time of the rover [104].

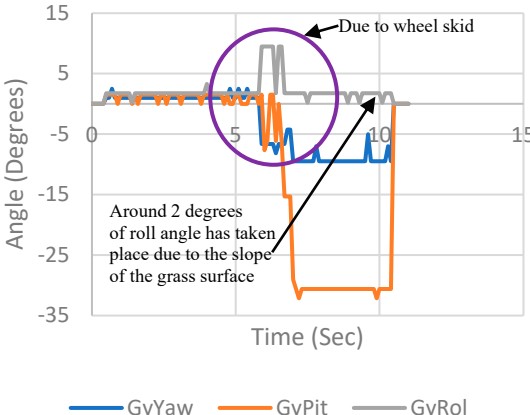

**Figure 17.** The actual yaw angle, pitch angle and roll angle vs. the time of the rover [104].

Figure 18 shows that the proposed dynamic metaheuristic FLC mechanism improved the rover's performance by coordinating all four wheel speeds to an optimal level.

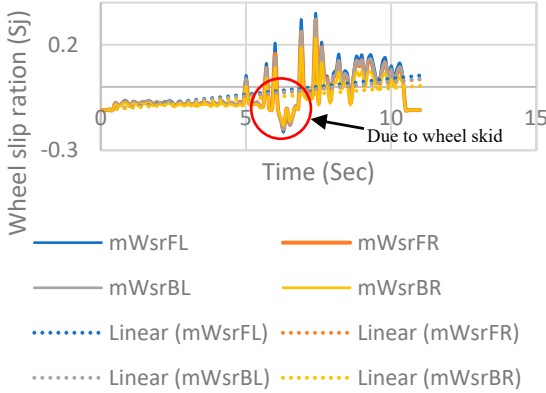

**Figure 18.** The wheel slip ratio ($S_j$) vs. time (Sec) for all the four wheels [104].

The "*mWsrFL*", "*mWsrFR*", "*mWsrBL*" and "*mWsrBR*" values in Figure 18 reflect the observed wheel slip ratio of the front-left, front-right, back-left and back-right wheels, respectively. Figure 18 depicts all of the wheel slip ratio graphs as having the same form. These events occurred as a result of the straight road test being done on a surface with an almost even kinetic friction coefficient. Furthermore, it demonstrates that the wheel slip ratio for all wheels was positive, and the proposed dynamic metaheuristic FLC kept the wheel slip ratio of all wheels within a range of less than 0.35.

Apart from the observed test results mentioned above to evaluate the performance of the proposed FLC, a comparative performance study between the proposed FLC and the

other typical control mechanisms is taken into consideration. However, there are few rover or vehicle verification studies in this research field, particularly studies on regulating the wheel slip independently while maintaining a fixed orientation on slippery roads under high-speed conditions (during acceleration and deceleration) via dynamic FLCs. In order to further verify the proposed metaheuristic FLCs' effect, studies involving a reference estimation model [111], an adaptive fuzzy type-2 control mechanism [112], $H_\infty$ with the Moore–Penrose theory [113], a torque distribution control [114], an electrical drive wheel speed using a machine learning approach [115], a longitudinal vehicle speed estimator based on fuzzy logic control [116], a torque vector control of a rear-wheel independent-drive (RWID) electric vehicle [117] and an anti-skid fuzzy PID control strategy for a four-wheel independent-drive electric vehicle (4WDIEV) [118] are selected to be compared and a validation simulation is carried out, except for [116], which is validated via both the simulation setup and hardware setup. A comprehensive performance comparison is shown in Table 12.

Therefore, it is evident that the suggested strategy has a good control impact on the wheel slip during high-speed acceleration or deceleration, with high feasibility for integration in the controller hardware.

**Table 12.** Comparative performance study between the proposed FLC model and typical dynamic models.

| Reference | Similar Research Works Recently Published | | | Compared Parameter(s) | | | Established Mechanism to Verify the Control Strategy | |
| --- | --- | --- | --- | --- | --- | --- | --- | --- |
| | Research Title and the Published Year | Control Strategy | The Controlled Physical Phenomenon | Controlled Parameter(s) of the Research Work | Controlled Parameter(s) of the *Proposed Metaheuristic FLC* | Advantages of the *Proposed Metaheuristic FLC System* | The Mechanism Used in the Research Work | The Mechanism Used in the *Proposed Metaheuristic FLC* |
| [111] | Research on Torque Distribution of Four-Wheel Independent Drive Off-Road Vehicle Based on PRLS Road Slope Estimation. *2021*. | PRLS Road Slope Estimation. | Wheel slip and orientation of the vehicle. | Wheel torque distribution. The maximum translational velocity was tested at around 25 km/h. The average wheel slip of all four wheels was 0.8 The maximum wheel torque was achieved at 2.4 kN·m | The angular speed of each wheel. The angular torque of each wheel. The desired orientation of the rover under high-speed conditions (sudden acceleration and deceleration). The rover had lateral stability, longitudinal stability and radial stability under high-speed conditions. The top recorded translational speed of the rover was approximately 90 km/h. The maximum translational (longitudinal) acceleration on wet grass slippery surface ($0.01 \leq \mu \leq 0.4$) was 3.4 ms$^{-2}$. The recorded wheel slip of the rover was less than 0.35. | The proposed metaheuristic FLC is independent of mathematical governing equation(s). | Hardware-in-the-loop real-time **simulation** and real vehicle tests. | The proposed dynamic metaheuristic FLC was tested via a **four-wheel independent-drive electric rover** model. Figures A9 and A10. |
| [112] | Adaptive Fuzzy Type-II Controller for Wheeled Mobile Robot with Disturbances and Wheel slips. *2021*. | Adaptive Fuzzy Type-II Control mechanism. | Wheel slip and trajectory follower. | Wheel torque distribution. The maximum recorded translational velocity was around 12.4 m/s. | | As the authors stated: "the control scheme is the complication in the mathematic proof". The proposed metaheuristic FLC system is independent of the system-governing equation(s). | Simulation setup. The authors have done a **simulation** with two types of reference trajectory: elliptical and Trifolium shapes. | |
| [113] | Control for four-wheel independently driven electric vehicles to improve steering performance using $H_\infty$. and Moore–Penrose theory. *2019*. | $H_\infty$ and Moore–Penrose theory. In this case, the authors developed a "logarithmic functional relationship between wheel cornering stiffness". | Wheel slip and orientation (yaw moment) of the vehicle. | Regulated the wheel cornering stiffness. Controlled the yaw moment of the 4WID EV. | | As the authors stated, the "decrease of adhesive force caused by the wear of the tyre could change the vehicle's dynamic property, and the design of a more robust controller adjusting to a varying vehicle system would bring some new challenges". This issue is not a problem for the proposed metaheuristic FLC because it is independent of the system-governing equation(s)/(mathematical model) | A **simulation** test setup has been established for the following three cornering stiffness ($S_{c\alpha}$) categories. Category 1: If $S_{c\alpha} > 1$ Category 2: If $S_{c\alpha} < 1$ Category 3: If $S_{c\alpha} = 1$ | |

**Table 12.** *Cont.*

| | Similar Research Works Recently Published | | | Compared Parameter(s) | | | Established Mechanism to Verify the Control Strategy | |
| --- | --- | --- | --- | --- | --- | --- | --- | --- |
| Reference | Research Title and the Published Year | Control Strategy | The Controlled Physical Phenomenon | Controlled Parameter(s) of the Research Work | Controlled Parameter(s) of the *Proposed Metaheuristic FLC* | Advantages of the *Proposed Metaheuristic FLC System* | The Mechanism Used in the Research Work | The Mechanism Used in the *Proposed Metaheuristic FLC* |
| [114] | A New Torque Distribution Control for Four-Wheel Independent-Drive Electric Vehicles. *2021*. | Torque distribution control. | Vehicle stability and handling performance, especially under extreme driving conditions. | Wheel torque distribution. Torque control was considered to achieve the desired yaw moment of the 4WIDEV. As the authors stated, they made "quicker and fuller use of lateral force to generate yaw moment and gained better vehicle stability". | | In this similar research work, an "ideal motion state estimator" was developed. However, when the mathematical model needed to become more realistic, all system information needed to be captured. The proposed FLC was tested in real-time through a hardware application (4WDI ER) and compared to similar research work. | HIL **simulation** has been utilized by the authors to verify the effectiveness of the proposed optimal torque distribution approach (two approaches have been considered). Approach 1: *Sine with Dwell:* The initial speed was set to 80 km/h. The friction coefficient was 0.8. Approach 2: *Double Lane Change* Closed-loop simulations have been conducted at a constant speed of 60 km/h. The friction coefficient was 0.8. | |
| [115] | A new application for fast prediction and protection of electrical drive wheel speed using machine learning methodology. *2022*. | Artificial neural network (ANN) coupled with particle swarm optimization (ANN-PSO). | Steering angle and steering ahead are achieved via an electronic differential control. | Angular velocity and wheel slip. The longitudinal forces, lateral forces and radial forces. The maximum recorded translational velocity was around 80 km/h. The maximum wheel torque was ≈ 138 N·m (total wheel torque) | | In this similar research work to stabilize a vehicle under uncertain conditions, the wheel speed or torque have to be regulated. Therefore, to achieve the desired electric current or voltage of the permanent magnet synchronize motor (PMSM), Lyapunov's stability analysis theory is taken into consideration. The proposed metaheuristic FLC is independent of non-linear mathematical (Lyapunov's, etc.) models. | The electric rear-wheel-drive PMSM speed regulation is **simulated** using the DTFC command. | |
| [116] | A Novel Longitudinal Speed Estimator for Four-Wheel Slip in Snowy Conditions. *2021*. | Longitudinal vehicle speed estimator based on fuzzy logic control. | Wheel angular velocity/torque and wheel slip. | Angular velocity and wheel slip. The translational velocity of the vehicle. Longitudinal acceleration of the vehicle. | | The authors stated in similar research work that "the estimated result is *not accurate in high-slip conditions*". However, when considering the proposed FLC mechanism, the observed test results show that the controller performed at an admirable level. The maximum translational speed of the rover is approximately 90 km/h, while synchronizing the all-wheel speed to achieve a fixed orientation. The average kinetic friction coefficient is around 0.1. Therefore, the proposed FLC has the ability to perform well under high wheel slip conditions. | **Experimental** and **simulation** tests have been carried out. Three driving condition cases have been taken into consideration. 1st Case: No wheel Slip. 2nd Case: At least one-wheel slips. 3rd Case: All four wheels slip. | |
| [117] | Torque Vectoring Control of RWID Electric Vehicle for Reducing Driving-Wheel Slippage Energy Dissipation in Cornering. *2021*. | Vector control mechanism. | Wheel angular velocity/torque and wheel slip. | Longitudinal linear stiffness of each driving wheel. The initial differential torque. Tire slippage energy dissipation. Acceleration slip regulation (ASR). | | | **Simulations** of typical maneuvering have been considered. | |

**Table 12.** *Cont.*

| Reference | Similar Research Works Recently Published | | | Compared Parameter(s) | | | Established Mechanism to Verify the Control Strategy | |
| --- | --- | --- | --- | --- | --- | --- | --- | --- |
| | Research Title and the Published Year | Control Strategy | The Controlled Physical Phenomenon | Controlled Parameter(s) of the Research Work | Controlled Parameter(s) of the *Proposed Metaheuristic FLC* | Advantages of the *Proposed Metaheuristic FLC System* | The Mechanism Used in the Research Work | The Mechanism Used in the *Proposed Metaheuristic FLC* |
| [118] | Research on Anti-Skid Control Strategy for Four-Wheel Independent Drive Electric Vehicle, *2021*. | Fuzzy PID Control strategy (Artificial Intelligence and classical control-based control strategy). | Anti-skid control. Wheel slip rate in real time. | Angular velocity and wheel slip. The maximum electric vehicle driving translational velocity was around 10 km/h. The driving torque of each independent driving wheel was 500 N·m. | | The authors stated that "he entire road surface identification process is in line with the *assumptions*". The proposed FLC has the ability to compensate for unexpected disturbances. | Based on Carsim and MAT-LAB/Simulink, the vehicle dynamics model, tire model and driving anti-skid control **simulation** model(s) have been established. | |

## 10. Conclusions

Table 1 expresses evidently that when controlling a sophisticated physical phenomenon via an FLC, hundreds of sophisticated fuzzy rules have to be implemented to build a realistic FLC. However, even in such a fuzzy inference mechanism, tuning the fuzzy rules is also a crucial issue. Moreover, the current studies on higher-order types of FLS, particularly the designed and developed applications of the interval type-2 fuzzy logic, have increased significantly because of the ability to compensate for uncertain conditions. However, there are concerns among the researchers due to the complexity of designing and constructing interval type-2 fuzzy controllers, which contain more parameters than their type-1 counterparts, which causes greater computational complexity and overhead issues.

Therefore, the motivation for this research is to design and develop a dynamic metaheuristic algorithm to automatically optimize the fuzzification, defuzzification and fuzzy reasoning processes by integrating the features of type-2 FLCs to give more accurate control results under uncertain conditions. This allows researchers to quickly develop more accurate fuzzy controllers.

However, to identify the performance of the developed dynamic metaheuristic FLC, as a piece of non-linear plant, a four-wheel independent-drive electric rover was taken into consideration. Moreover, compared to previous work and approaches to a similar research problem, the proposed dynamic metaheuristic FLC with a self-optimization mechanism has the ability to improve the rover's stability during its fastest acceleration and deceleration phases in slippery road conditions. Figure 1 shows the proposed dynamic metaheuristic FLC (*controller A*) along with the designed and constructed steering FLC (S-FLC (*controller B*)) and the differential FLC (D-FLC (*controller C*)), which is capable of driving the rover at a notable peak translational velocity of 90 km/h (55 mph). However, during testing (on slippery roads), the rover's translational velocity was reduced to 45 km/h (28 mph). Furthermore, the proposed dynamic metaheuristic FLC with the D-FLC kept the wheel slip ratio of all four wheels within an admirable range of less than 0.35 on a wet grass surface with an average friction coefficient $\mu_j$ of 0.12.

For subsequent investigations, future work should be carried out to identify the efficiency or performance characteristics of the yaw moment generation for independent wheel torque distribution via the proposed dynamic metaheuristic FLC. Moreover, under high-speed conditions, future studies should investigate how the proposed FLC compensates for negotiating curves while maintaining the desired lateral forces to enhance the rover's stability.

**Author Contributions:** Conceptualization, H.R.J. and W.R.d.M.; Methodology, H.R.J.; Software, H.R.J.; Validation, H.R.J.; Formal analysis, H.R.J.; Investigation, H.R.J., W.R.d.M. and S.C.M.; Data curation, H.R.J.; Writing-Original draft preparation, H.R.J., Writing-review and editing, W.R.d.M. and S.C.M.; Visualization, H.R.J.; Supervision, W.R.d.M. and S.C.M. All authors have read and agreed to the published version of the manuscript.

**Funding:** This research received no external funding.

**Institutional Review Board Statement:** Not applicable.

**Informed Consent Statement:** Not applicable.

**Data Availability Statement:** Data are contained within the article.

**Conflicts of Interest:** The authors declare no conflict of interest.

## Appendix A

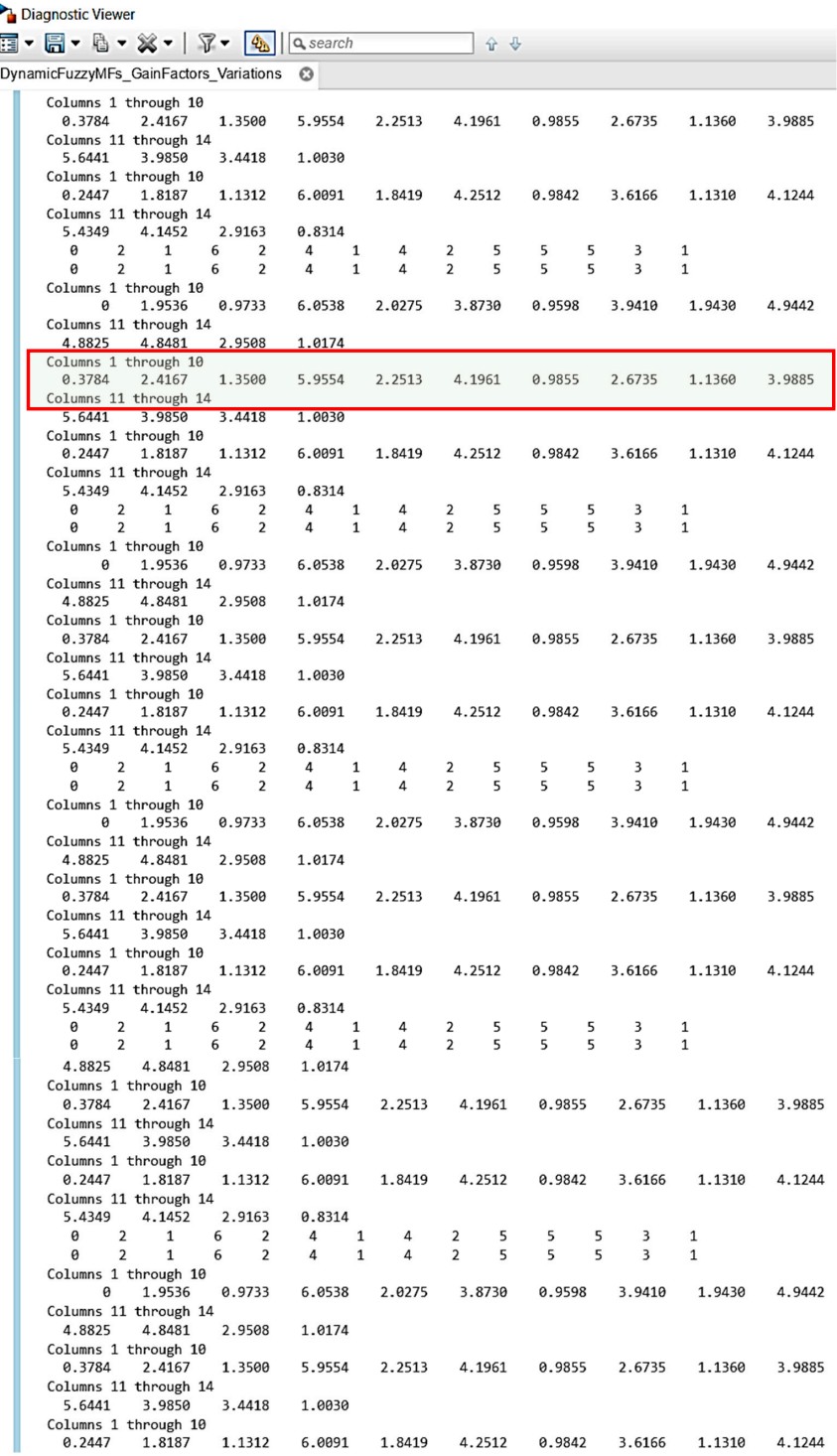

**Figure A1.** The dynamic variations of the fourteen (14) common membership gain factors.

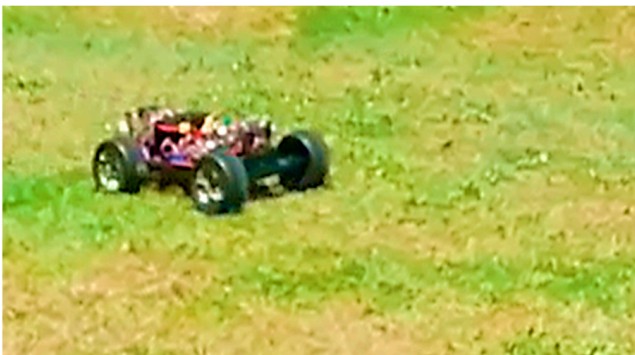

**Figure A2.** The straight road test on a slippery wet grass surface (captured when the speed of the rover was around 42.8 km/h.).

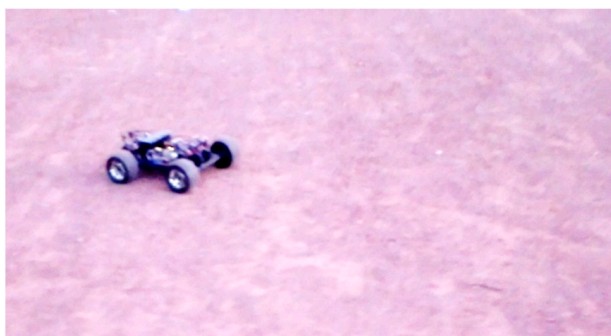

**Figure A3.** Top recorded speed while running on a gravel soil surface (as shown in Figure A8, the top recorded speed was around 90 km/h).

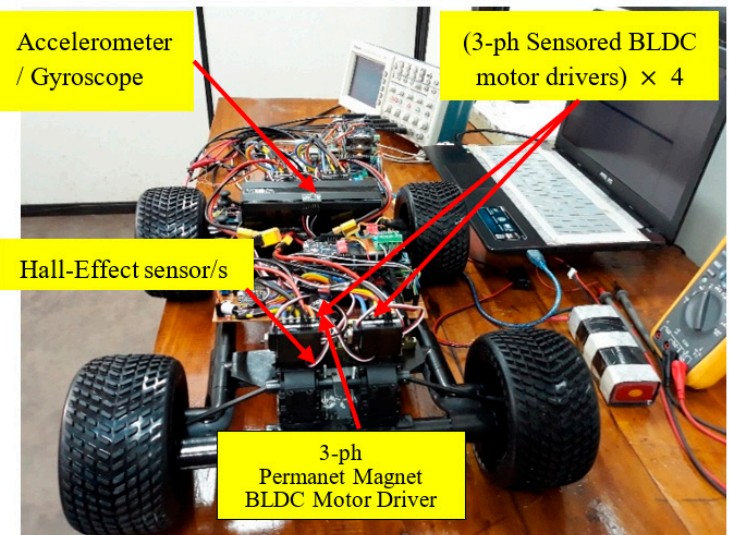

**Figure A4.** The designed and developed 4WD electric rover with the TSK-PSO-FLC.

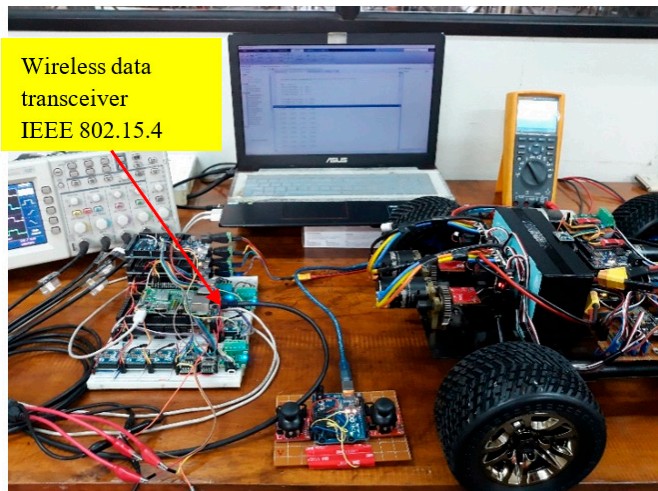

**Figure A5.** The developed wireless transceiver with a system database (up to 2 km).

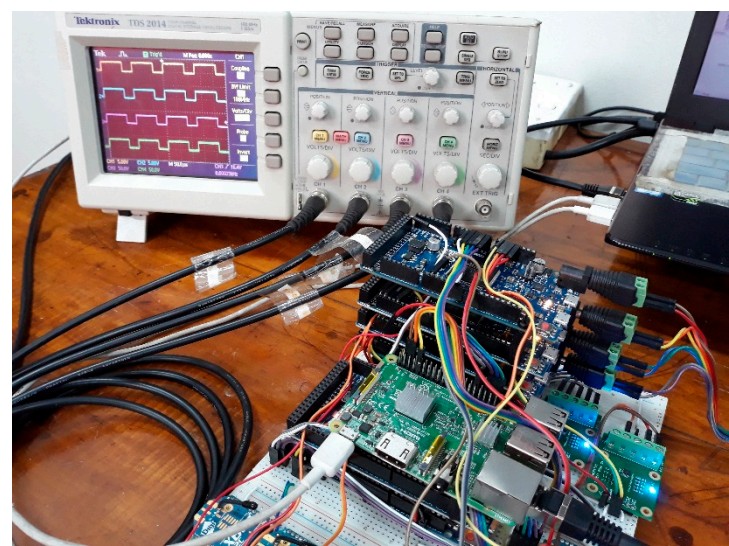

**Figure A6.** The synchronized four-motor drive signal for every sampling instant "*k*".

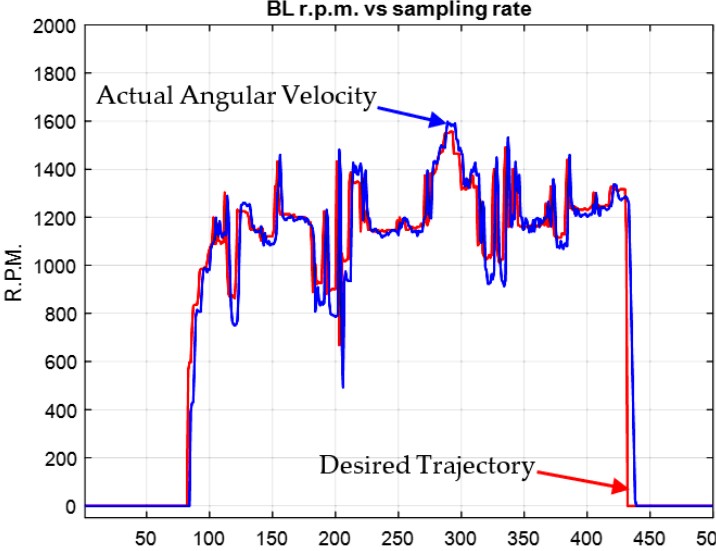

**Figure A7.** The back-left tire response for the desired input trajectory (50 samples/s).

Figures A4–A6 show the designed and developed experimental test setup that was utilized to control the electric rover while observing data wirelessly in the designed and developed database (via IEEE 802.15.04 protocol). Figure A7 shows the desired angular velocity trajectory generated via the TSK-PSO-FLC and the observed actual angular velocity of the back-left wheel during the laboratory test.

**Table A1.** The physical parameters of the electric rover.

| Physical Parameter | Amount with Units |
|---|---|
| Rover Width ($W$) | 0.415 m |
| Rover Height ($H$) (Ground clearance) | 0.06 m |
| Rover Length ($L$) | 0.465 m |
| Diameter of a wheel | 0.13 m |
| Weight of the rover body ($M_B$) | 3.288 kg |
| Weight of a wheel ($M_{Wh}$) | 0.064 kg |
| Total weight of the rover ($M_R$) | 5.066 kg |

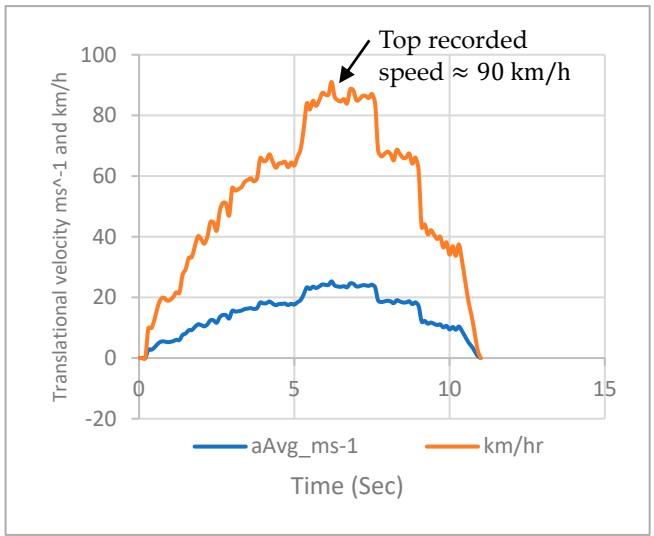

**Figure A8.** Top recorded speed of the rover (has not been discussed in this paper).

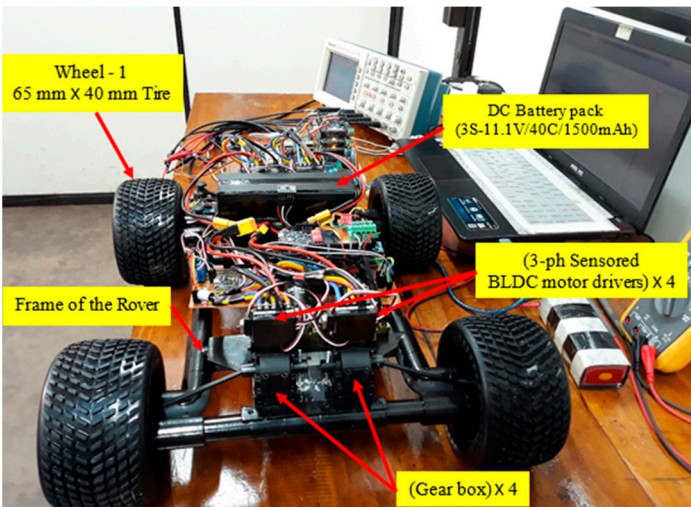

**Figure A9.** The front view of the electric rover.

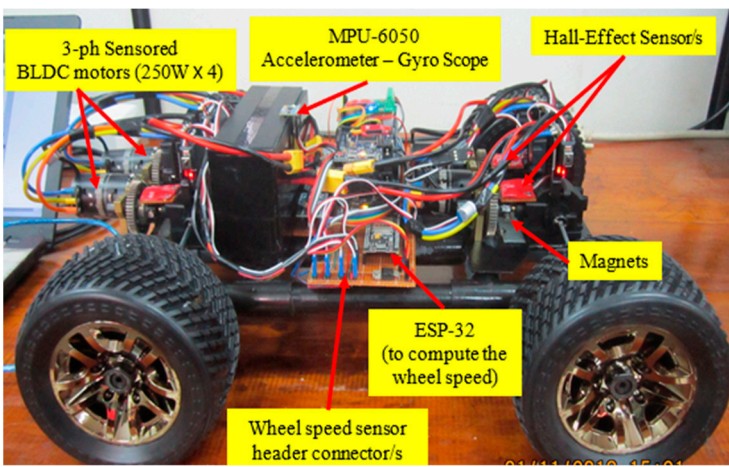

**Figure A10.** The side view of the electric rover.

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
