# Peer review of "Real-Time Metaheuristic Algorithm for Dynamic Fuzzification, De-Fuzzification and Fuzzy Reasoning Processes"

_applsci, doi:10.3390/app12168242_

Round 1

Reviewer 1 Report

This paper presents a Fuzzy Logic Controller design based on dynamic Fuzzification, Defuzzification, and Fuzzy reasoning through the Particle Swarm Optimization (PSO) approach for tunning. The proposed model is applied to four independent wheels electric cars.

The paper approach and the goal are described and developed clearly throughout the paper. However, I would like to advise some changes and additions that should help improve the paper's impact and soundness.

1. The introduction section and Table 1 try to set up state of the art related to the Fuzzy Inference System development. However, a paragraph should be included highlighting the difference between the current state of the art in this area and the design proposed by the current paper.

2. The Table 1 Caption does not make sense. In this context, it is possible to guess the real meaning of this caption; however, by itself, this caption is not clear. (Please, rephrase this sentence)

3. Before section 2. It should include a paragraph enumerating and explaining the contributions of the proposed system. Additionally, the paper's contributions should be contrasted with the state-of-the-art technics. 

4. The PSO version expressed by Eq (18 and 19) is the classical and basic PSO formulation. This version is widely known because of its incapacity to overcome local minima.

It should include a section to explain how the PSO is modified to overcome the local minima problem.

5. Figure 9 is not needed

6. For the Results and Discussion, a comparative performance study between the proposed model and the classical and typical dynamic model used for state of the art, should be included. This Comparative analysis should have a qualitative and quantitative approach.  

Additionally, the proposed model should include some strategies that demonstrate how the proposed model avoids being stuck in local minima. Finally, this strategy should be evaluated and compared with the classical one to demonstrate the proposed model's improvement. 

Grammatical Issues: The paper has some grammatical issues (no typos); some punctuation marks are wrong allocated. These errors crack the sentence sense.

Example: Section 5. 

The sentence before the last one. "Therefore, instead of developing such a sophisticated fuzzy inference mechanism, to overcome these issues the proposed dynamic metaheuristic FLC, is capable of dynamically optimizing the fuzzy reasoning process."

Correct use of the punctuation marks "Therefore, instead of developing such a sophisticated fuzzy inference mechanism to overcome these issues, the proposed dynamic metaheuristic FLC is capable of dynamically optimizing the fuzzy reasoning process."

The complete paper is full of these mistakes; I would like to advise you to check the entire paper's grammar and fix the improper uses of the punctuation marks.

In general, I would like to advise the Authors to redirect the paper's approach from the case report to the research results report; it's aimed at increasing the paper's potential impact and improving the contribution to state of the art. 

Author Response

Manuscript ID: applsci-1817106

Title: Real-Time Metaheuristic Algorithm for Dynamic Fuzzification, De-fuzzification and Fuzzy Reasoning Process

Authors: Subhas Chandra Mukhopadhyay *, Hasitha R Jayetileke, WR De Mel

Revised: 03rd August 2022

Revised Manuscript According to the Referees' Comments

Reviewer#1

1st Comment:

The introduction section and Table 1 try to set up state of the art related to the Fuzzy Inference System development. However, a paragraph should be included highlighting the difference between the current state of the art in this area and the design proposed by the current paper.

Ans: (On page 6)

However, the level of uncertainty in a system could be minimized by employing interval type-2 fuzzy logic, which has stronger capabilities to handle uncertainties by modelling the vagueness and unpredictability of information [44–47]. This is because-of, with the growth of type-2 FLS uncertainty could be directly integrated into fuzzy sets which have been described in Section 6. Furthermore, in the last three years of study on high-er-order types of FLS particularly, the designed and developed application of interval type-2 fuzzy logic has increased significantly [48–54]. These type-2 based FLS/applications could be identified in Artificial Intelligence (AI) [55–59], Adaptive Control [60–66], Electric Motor Control [67–72], Internet of Things (IoT) [73–77], Digital Image Processing [78–84] and others [85–87]. Of course, the application of interval type-2 fuzzy logic in the domain of control has recently attracted a lot of attention due to its better performance under uncertain conditions. The fundamental issue, however, is the complexity of designing and constructing interval type-2 fuzzy controllers, which contain more parameters than their type-1 counterparts therefore this makes a greater computational complexity, and overhead issues [88–99]. Therefore, several efforts were made to reduce the complexity of generalized interval type-2 fuzzy logic systems; for example, Samui and Samarjit [100] have published Neural Network (NN) based tuning mechanism and Cagri and Tufan [101] have put forward by developing differential flatness-based controller which both enable the computation with Generalized Type-2 FLS (GT2FLS). However, no general design strategy for finding the optimal type-2 fuzzy model has been proposed yet [102].  

2nd Comment:

The Table 1 Caption does not make sense. In this context, it is possible to guess the real meaning of this caption; however, by itself, this caption is not clear. (Please, rephrase this sentence)

Ans: (On page 3)

Table 1. A general summary of the current state of the research area that shows sophisticated development of fuzzy Input/Output membership functions (shown in the 4th and 5th column) and especially when developing the FIM (6th column shows evidence for the development of a large number of fuzzy rules) to compensate for sophisticated physical phenomena.

3rd Comment:

Before section 2. It should include a paragraph enumerating and explaining the contributions of the proposed system. Additionally, the paper's contributions should be contrasted with the state-of-the-art technics.

Ans. (on page 6)

The main contribution of this paper is to design and develop a robust FLC which enables researchers to rapidly develop more realistic fuzzy controller(s). Therefore, to overcome the problems/drawbacks from the above-mentioned review of previous studies the main advantages of this study are:

  • Evaluated how the Type-2 FLS does are more capable of performing under uncertain condition(s) and designed and developed a mechanism to integrate it into the proposed Type-1 FLC.
  • Designed and developed an adaptive metaheuristic FIM for Type-1 FLS to overcome the problems that are currently faced when developing realistic fuzzy rules (as shown in Table 1, column 6).
  • Designed and developed a fuzzification and de-fuzzification mechanism while integrating the features that were abstracted from Type-2 FLS into Type-1 FLS.
  • A real-time dynamic metaheuristic algorithm to automatically optimize all the above-mentioned processes related to dynamic fuzzification, de-fuzzification and fuzzy reasoning process have been designed and developed.
  • To examine the performance of the proposed controller as a complex physical phenomenon, a four-wheel independent drive electric rover has been designed and developed to regulate the wheel slip (under high-speed conditions on slippery roads).

4th Comment:

The PSO version expressed by Eq (18 and 19) is the classical and basic PSO formulation. This version is widely known because of its incapacity to overcome local minima.

It should include a section to explain how the PSO is modified to overcome the local minima problem.

Ans. (On page 14)

However, it has been noticed that during the programme execution process once the best particle in the global traps in a local minimum, all particles followed that particle and are trapped in the same local minimum. In that case, once they are trapped in local minima, to overcome this issue the parameter values in (19) have been switched via a separately dedicated Takagi-Sugeno-Kang (TSK) FLC. The design and development of the TSK-FLC have been discussed in detail in Section 7. Finally, this mechanism enabled the particles to find a completely new solution set for the next generation.

5th Comment:

Figure 9 is not needed.

Ans. (On page 41)

Removed it from the article's main body and included it in Appendix A.

6th Comment:

For the Results and Discussion, a comparative performance study between the proposed model and the classical and typical dynamic model used for state of the art, should be included. This Comparative analysis should have a qualitative and quantitative approach.

Ans. (on page 28)

Apart from the observed test results mentioned above to evaluate the performance of the proposed FLC a comparative performance study between the proposed FLC and the other typical control mechanisms has been taken into consideration. However, there are few rover/vehicle verifications in this research field particularly on regulating the wheel slip independently while maintaining a fixed orientation on slippery roads under high-speed conditions (during acceleration and deceleration) via dynamic FLCs. In order to further verify the proposed metaheuristic FLCs’ effect, Ref. [111] with reference estimation model, Ref. [112] with an adaptive fuzzy type-2 control mechanism, Ref. [113]  an Moore–Penrose theory, Ref. [114] torque distribution control, Ref. [115] electrical drive wheel speed using machine learning approach, Ref. [116] longitudinal vehicle speed estimator based on fuzzy logic control Ref. [117] torque vectoring control of Rear-Wheel Independent-Drive (RWID) electric vehicle and Ref. [118] anti-skid fuzzy-PID control strategy for Four-Wheel Independent Drive Electric Vehicle (4WDIEV) are selected to be compared, that carried out simulation validation except Ref. [116] which has been validated via both simulation setup and hardware setup. The comprehensive performance comparison has been shown in Table 12.

Therefore, it is evident that the suggested strategy has a good control impact on wheel slip during high-speed acceleration/deceleration with high feasibility in the controller hardware.  

Table 12. Comparative performance study between the proposed FLC model and typical dynamic model(s)

Reference

Similar research works recently        published

Compared parameter(s)

Advantages of the proposed metaheuristic FLC system

Established mechanism to verify the control strategy

Research title and the published year

Control strategy

The controlled    physical                phenomenon

Controlled parameter(s) of the similar research work

Controlled parameter(s) of the proposed metaheuristic FLC

The mechanism used in the similar research work

The mechanism used in the proposed metaheuristic FLC

[111]

Research on Torque Distribution of Four-Wheel Independent

Drive Off-Road Vehicle Based on PRLS Road Slope Estimation.  2021.

PRLS Road Slope Estimation.

Wheel slip and the orientation of the vehicle.

Wheel torque distribution.

The maximum translational velocity has been tested at around 25 km/hr.

The average wheel slip of all four wheels is: 0.8

The maximum wheel torque has been achieved: 2.4 kN.m

The angular speed of each wheel.

Angular torque of each wheel.

Desired orientation of the rover under high-speed conditions (Sudden acceleration and deceleration).

The rover has lateral stability, longitudinal stability and radial stability under high-speed conditions.

The top recorded translational speed of the rover is approximately 90 km/hr.

The maximum translational (longitudinal) acceleration on wet grass slippery surface () is 3.4 .

The recorded wheel slip of the rover is less than 0.35. 

The proposed metaheuristic  FLC is independent of mathematical governing equation(s).  

Hardware-in-the-loop real-time simulation and real vehicle tests.

The proposed dynamic metaheuristic FLC has been tested via a Four-wheel independent drive electric rover model. Figure B8 and Figure B9.

[112]

Adaptive Fuzzy Type-II Controller for Wheeled Mobile

Robot with Disturbances and Wheel slips. 2021.

Adaptive Fuzzy Type-II Control mechanism.

Wheel slip and the trajectory follower.

Wheel torque distribution.

The maximum recorded translational velocity is around 12.4 m/s.

As the authors have stated: “the control scheme is the complication in the mathematic proof.”

The proposed metaheuristic FLC system is independent of the system governing equation(s).

Simulation setup.

The authors have done a simulation with two types of reference

trajectory: elliptical and Trifolium shapes.

[113]

Control for four-wheel independently

driven electric vehicles to improve

steering performance using  and

Moore–Penrose theory.

2019.

H_∞ and

Moore–Penrose theory.

In that case, authors have developed a “logarithmic functional relationship between wheel cornering stiffness”.

Wheel slip and the orientation (yaw moment) of the vehicle.

Regulating the wheel cornering stiffness.

Controlled yaw moment of the 4WID EV.

As the authors have stated “decrease of adhesive force caused by the

wear of tyre could change the vehicle's dynamic property, and the design of a more robust controller adjusting to a varying vehicle system would bring some new

challenges.”

This issue is not a problem for the proposed metaheuristic FLC because of independent from system governing equation(s)/(mathematical model)

A simulation test setup has been established for the following three cornering stiffness () categories.

Category 1:

If

Category 2:

If

Category 3:

If

[114]

A New Torque Distribution Control for Four-Wheel

Independent-Drive Electric Vehicles. 2021.

Torque distribution control.

Vehicle stability and handling performance, especially under

extreme driving conditions.

Wheel torque distribution.

Torque control has been considered to achieve the desired yaw moment of the 4WIDEV.

As the authors stated made “quicker and fuller use of lateral force to generate yaw

moment and gained better vehicle stability”.

In this similar research work “Ideal motion state estimator” has been developed. However, when a mathematical model needed to become more realistic all the system information needs to be captured.

The proposed FLC has been tested in real-time through a hardware application (4WDI ER) compared to similar research work.

HIL simulation has been utilized by the authors to verify the effectiveness of the proposed optimal

torque distribution approach (two approaches have been considered).

Approach 1:

Sine with Dwell:

The initial speed was set to 80 km/h.

The friction coefficient was 0.8.

Approach 2:

Double Lane Change

Closed-loop simulations have been conducted at a constant speed of 60 km/h.

The friction

coefficient was 0.8.

[115]

A new application for fast prediction and protection of electrical

drive wheel speed using machine learning methodology. 2022.

Artificial neural network (ANN)

coupled with particle swarm optimization (ANN-PSO).

Steering angle and the steering ahead are achieved via an electronic differential control.

Angular velocity and the wheel slip.

The longitudinal forces, lateral forces and radial forces.

The maximum recorded translational velocity is around 80 km/hr.

The maximum wheel torque has been achieved  138 N.m (Total wheel torque)

In this similar research work to stabilize the vehicle under uncertain conditions, the wheel speed/torque has to be regulated. Therefore, to achieve the desired electric current/voltage of the Permanent Magnet Synchronize Motor (PMSM) Lyapunov's stability analysis has been taken into consideration.

The proposed metaheuristic FLC is independent of non-linear mathematical (Lyapunov's etc…) models.

The electric rear-wheel drive PMSM speed regulation is simulated using the DTFC

command.

[116]

A Novel Longitudinal Speed Estimator for Four-Wheel Slip in

Snowy Conditions. 2021.

Longitudinal vehicle speed estimator based on fuzzy logic control. 

Wheel angular velocity/torque and the wheel slip.

Angular velocity and wheel slip

The translational velocity of the vehicle.

Longitudinal acceleration of the vehicle.

As per the authors have stated in similar research work “the estimated result is not accurate in high slip conditions”.

However, when considering the proposed FLC mechanism the observed test results show that the controller has been performed at an admirable level.

The maximum translational speed of the rover is approximately 90km/hr while synchronizing the all-wheel speed to achieve a fixed orientation.

The average kinetic friction coefficient is around 0.1. Therefore, the proposed FLC has the ability to perform well under high wheel slip conditions.

Experimental and simulation tests have been carried out.

Three driving condition cases have been taken into consideration.

1st Case:

No wheel

Slip.

2nd Case:

At least one-wheel slips.

3rd Case:

All four wheels slip.

[117]

Torque Vectoring Control of RWID Electric Vehicle for

Reducing Driving-Wheel Slippage Energy Dissipation

in Cornering. 2021.

Vectoring control mechanism.

Wheel angular velocity/torque and the wheel slip.

Longitudinal linear stiffness of each driving wheel.

The initial differential torque.

Tire slippage energy dissipation.

Acceleration slip regulation (ASR).

Simulations of typical manoeuvring have been considered.

[118]

Research on Anti-Skid Control Strategy for Four-Wheel

Independent Drive Electric Vehicle, 2021.

Fuzzy PID

Control strategy.

(Artificial Intelligence and classical control-based control strategy).

Anti-skid control.

Wheel slip rate in real-time.

Angular velocity and wheel slip.

The maximum electric vehicle driving translational velocity is around 10 km/h.

The driving torque of each independent driving

the wheel is 500 N·m.

As per the authors have been stated “The entire road surface identification process

is in line with the assumptions.”

The proposed FLC has the ability to compensate for unexpected disturbances.

Based on Carsim and MATLAB/Simulink, the vehicle dynamics model, tire model

and driving anti-skid control simulation model(s) have been established.

General Comments:

Additionally, the proposed model should include some strategies that demonstrate how the proposed model avoids being stuck in local minima. Finally, this strategy should be evaluated and compared with the classical one to demonstrate the proposed model's improvement.

Ans.

Answered when answering for the 4th Comment.

Grammatical Issues:

The paper has some grammatical issues (no typos); some punctuation marks are wrong allocated. These errors crack the sentence sense.

Example: Section 5.

The sentence before the last one. "Therefore, instead of developing such a sophisticated fuzzy inference mechanism, to overcome these issues the proposed dynamic metaheuristic FLC, is capable of dynamically optimizing the fuzzy reasoning process."

Correct use of the punctuation marks "Therefore, instead of developing such a sophisticated fuzzy inference mechanism to overcome these issues, the proposed dynamic metaheuristic FLC is capable of dynamically optimizing the fuzzy reasoning process."

The complete paper is full of these mistakes; I would like to advise you to check the entire paper's grammar and fix the improper uses of the punctuation marks.

Ans.

Checked the entire paper and corrected all the mistakes.

In general, I would like to advise the Authors to redirect the paper's approach from the case report to the research results report; it's aimed at increasing the paper's potential impact and improving the contribution to state of the art.

Ans.

Increased the paper’s potential impact through the paper and added more references for the justifications/evidence.

Reviewer 2 Report

Generally speaking, the problem proposed in this paper is interesting, and the obtained results are promising and correct.

The references in this paper are appropriate and most of the references cited by the author are recent ones.

The proposed methodology is significant, and the disclosed results are useful. I think that the paper is acceptable. 

It is enough to support the research in this paper. Nevertheless, there are some problems that should be addressed.

The results and conclusions of the paper are incomplete. Furthermore, future work should be provided at the end of the conclusion section.

Author Response

Reviewer#2

Comments:

The references in this paper are appropriate and most of the references cited by the author are recent ones.

The proposed methodology is significant, and the disclosed results are useful. I think that the paper is acceptable.

It is enough to support the research in this paper. Nevertheless, there are some problems that should be addressed.

The results and conclusions of the paper are incomplete. Furthermore, future work should be provided at the end of the conclusion section.

Ans.

Answered. See the answer that was given for the 6th Comment, Reviewer#1.

Reviewer 3 Report

I appreciate the effort made to document and develop this article proposal.

The foray into the stage of knowledge is based on a reasonable number of bibliographic landmarks.

I also appreciate the effort made to achieve the empirical application part of the article proposal.

In conclusion, I appreciate the entire effort and recommend the publication of this article proposal.

Author Response

Reviewer#3

Comment:

I appreciate the effort made to document and develop this article proposal.

The foray into the stage of knowledge is based on a reasonable number of bibliographic landmarks.

I also appreciate the effort made to achieve the empirical application part of the article proposal.

In conclusion, I appreciate the entire effort and recommend the publication of this article proposal.

Ans.

No technical comments to be addressed.

Reviewer 4 Report

applsci-1817106-peer-review-v1

Real-Time Metaheuristic Algorithm for Dynamic Fuzzification, De-fuzzification and Fuzzy Reasoning Process.

This paper introduces a real-time metaheuristic algorithm for dynamic fuzzification, de-fuzzification and fuzzy reasoning process. The model is formulated by dynamic fuzzification, de-fuzzification and fuzzy reasoning process. The paper contains a literature survey, some preliminaries (Definitions and notations). Problem formulation starts with a typically extended dynamic fuzzification, de-fuzzification and fuzzy reasoning process model introduced by Jayetileke et al. [45] and some definitions are given. Finally, numerical experimentation is presented.

I feel that although this may be a useful exercise, the paper does not come up to the international standards for publication. There is hardly any original contribution in this paper and it adds very little to the published literature. Control parameters of applied algorithms should be reported for each study case.

In view of this, I do not recommend this paper for publication in the applied sciences journal.

***

Author Response

Reviewer#4

Comment:

Real-Time Metaheuristic Algorithm for Dynamic Fuzzification, De-fuzzification and Fuzzy Reasoning Process.

This paper introduces a real-time metaheuristic algorithm for dynamic fuzzification, de-fuzzification and fuzzy reasoning process. The model is formulated by dynamic fuzzification, de-fuzzification and fuzzy reasoning process. The paper contains a literature survey, some preliminaries (Definitions and notations). Problem formulation starts with a typically extended dynamic fuzzification, de-fuzzification and fuzzy reasoning process model introduced by Jayetileke et al. [45] and some definitions are given. Finally, numerical experimentation is presented.

I feel that although this may be a useful exercise, the paper does not come up to the international standards for publication. There is hardly any original contribution in this paper, and it adds very little to the published literature. Control parameters of applied algorithms should be reported for each study case. In view of this, I do not recommend this paper for publication in the applied sciences journal.

Ans.

Added more publications to highlight the difference between the current state of this particular research field and the designed and developed proposed fuzzy logic controller.

Moreover, please be kind enough to refer to the answers provided for the first reviewer.

The reference [45], “Jayetileke et al.” before the revision and currently correspond to the reference [105] which is the same work of the authors.

Round 2

Reviewer 1 Report

Thank you for considering the comments and required additions.

Reviewer 4 Report

No comment